# Sobolev Acceleration and Statistical Optimality for Learning Elliptic Equations via Gradient Descent

**Yiping Lu**
ICME
Stanford University
Stanford, CA 94305
yplu@stanford.edu

**Jose Blanchet**
Management Science and Engineering
Stanford University
Stanford, CA 94305
jose.blanchet@stanford.edu

**Lexing Ying**
Department of Mathematics
Stanford University
Stanford, CA 94305
lexing@stanford.edu

## Abstract

In this paper, we study the statistical limits in terms of Sobolev norms of gradient descent for solving inverse problems from randomly sampled noisy observations using a general class of objective functions. Our class of objective functions includes Sobolev training for kernel regression, Deep Ritz Methods (DRM), and Physics Informed Neural Networks (PINN) for solving elliptic partial differential equations (PDEs) as special cases. We consider a potentially infinite-dimensional parameterization of our model using a suitable Reproducing Kernel Hilbert Space and a continuous parameterization of problem hardness through the definition of kernel integral operators. We prove that gradient descent over this objective function can also achieve statistical optimality and the optimal number of passes over the data increases with sample size. Based on our theory, we explain an implicit acceleration of using a Sobolev norm as the objective function for training, inferring that the optimal number of epochs of DRM becomes larger than the number of PINN when both the data size and the hardness of tasks increase, although both DRM and PINN can achieve statistical optimality.

## 1 Introduction

Several learning based methods for solving inverse problems have been proposed recently with state-of-the-art performance across a wide range of tasks, including medical image reconstruction [1], inverse scattering [2] and 3D reconstruction [3]. In this paper, we study the statistical limit of machine learning methods for solving inverse problems. To be specific, we consider the problem of reconstructing a function from random sampled observations with statistical noise in measurements. We apply gradient descent to a general class of objective functions for the reconstruction. When the observations are the direct observations of the function, the problem is non-parametric function estimation [4, 5]. The observations may also come from certain physical laws described by a partial differential equation (PDE)[6, 7]. Formally, we aim to reconstruct a function $f^*$ based on independently sampled data set $D = \{(x_i, y_i)\}_{i=1}^n$ from an unknown distribution $P$ on $\mathcal{X} \times \mathcal{Y}$, where $y_i$ is the noisy measurement of $u^*$ though a measurement procedure $\mathcal{A}$. For simplicity, we assume $\mathcal{A}$ is self-adjoint in this paper. The conditional mean function $f^*(x) = \mathbb{E}_P(Y|X = x)$ is the ground truth function for observation of $u^*$ through the measurement procedure $\mathcal{A}$, i.e. $f^* = \mathcal{A}u^*$. To solve this problem, we consider gradient descending over the following general class of objective function

$$\hat{u} = \arg\min_{u \in \mathcal{H}} \mathbb{E}_{\mathbb{P}_n(x,y)} \frac{1}{2} \langle u(x), \mathcal{A}_1 u(x) \rangle - \langle y, \mathcal{A}_2 u(x) \rangle,$$

where $\mathbb{P}_n = \frac{1}{n}\sum_{i=1}^n \delta(x_i, y_i)$ is the empirical distribution, $\mathcal{H}$ is a reproducing kernel Hilbert space (RKHS) and $\mathcal{A}_i, i = 1, 2$ are two self-adjoint operators that satisfy $\mathcal{A}_1 = \mathcal{A}\mathcal{A}_2$. In Section 2, we show that several algorithms, including kernel regression [4, 8] via Sobolev training [9, 10, 11] and solving PDEs via machine learning based algorithm, [12, 13, 2, 14] can be considered as special cases of this formulation.

36th Conference on Neural Information Processing Systems (NeurIPS 2022).

Recent works [15, 16] have considered the statistical limit of learning of elliptic inverse problem, *i.e.* how many observation of the right hand side function of an elliptic PDE are needed to reach a prescribed performance level. However, none of these papers consider computationally feasible methods for constructing such optimal estimators. In this paper, we consider the statistical optimality of gradient descent [17, 18, 19, 20], a successful and widely used algorithm in machine learning. We show that proper early stopped gradient descent can achieve information theoretical optimal convergence rate according to a continuous scale of suitable Hilbert norm (*i.e.* Sobolev norms[21, 22], detailed definition see Section 2).

We first prove that a properly early stopped gradient descent algorithm over the class of objective functions can achieve statistical optimality. At the same time, although any suitably early stopped gradient flow of the class of loss function can achieve statistical optimality according to our theory, we discover an acceleration effect of using a Sobolev norm as the loss function for kernel based machine learning algorithms. The implicit acceleration of the Sobolev loss function arises because a differential operator can enlarge the small eigenvalue of the kernel integral operator for high frequency functions, leading to better condition numbers and faster convergence in these eigenspaces while maintaining the statistical optimality. We justify our theoretical finding with several numerical experiments.

## 1.1 Related Works

**Machine Learning Based PDE Solver.** Partial differential equations (PDEs) are widely used in many disciplines of science and engineering and play a prominent role in modeling and forecasting the dynamics of multiphysics and multiscale systems. The recent breakthroughs in deep learning and the rapid development of sensors, computational power, and data storage in the past decade have drawn attention to numerically solving PDEs via machine learning methods [23, 24, 12, 25, 13, 2], especially in high dimensions where conventional methods become impractical. Based on the natural idea of representing solutions of PDEs by (deep) neural networks, different loss functions for solving PDEs are proposed. [25, 26] utilize the Feynman-Kac formulation which turns solving a PDE into a stochastic control problem. The work of [27] solves the weak formulations of PDEs via an adversarial network. In this paper, we focus on the convergence rate of the Deep Ritz Method (DRM) [14, 2] and the Physics-Informed neural network (PINN) [12, 13]. DRM [14, 2] utilizes the variational structure of the PDE, which is similar to the Ritz-Galerkin method in classical numerical analysis of PDEs, and trains a neural network to minimize the variational objective. PINN [12, 13] trains a neural network directly to minimize the residual of the PDE, i.e., using the strong form of the PDE. Theoretical convergence results for deep learning based PDE solvers has also received considerable attention recently. Specifically, [28, 29, 30, 31, 32, 33, 34] investigated the regularity of PDEs approximated by a neural network and [28, 35, 36, 37, 38] further provided generalization analyses. [15, 16, 39, 40] provided information theoretical optimal lower and upper bounds for solving PDEs from random samples. However, all these papers assume accessibility of the global solution of empirical loss minimization. In contrast, here we consider the gradient descent algorithm for learning the estimator. The most relevant work in connection to is [41], which considers a polynomial-time Langevin-type algorithms to sample from the posterior measure of the Bayesian inverse methods. Instead of considering the Bayesian setting, here we optimize on the un-regularized objective. However, the estimator is regularized via early stopping [42, 43, 44], *i.e.* we consider the statistical optimality of the implicit regularization effect of optimization algorithm. A concurrent paper [45] considered similar stochastic gradient descent approach for statistical inverse problem.

**Learning with kernel.** Supervised least square regression in RKHS has a long history and its generalization ability and mini-max optimality has been thoroughly studied [8, 46, 4, 47, 48]. Statistical optimality of early stopped (stochastic) gradient descent has been widely discussed in [42, 49, 50, 18, 17, 51, 52]. The convergence of least square regression in Sobolev norm has been discussed recently in [21, 22]. Recently, training neural networks with stochastic gradient descent in certain regimes has been found to be equivalent to kernel regression [53, 54, 55]. Gradient descent training of neural network in the kernel regime has been found optimal for a wide class of non-parametric functions with both early stopping regularization and ridge regression [56, 57].

## 1.2 Contribution

- We provide information theoretical lower bounds (Theorem 3.1) for a wide class of inverse problems, including the Sobolev learning rate [21] for the solution of elliptic inverse

problems. We also show that the previous lower bound [15, 16] for machine learning solving elliptic equations can be considered as a special case of our lower bound.

- We provide a proof of statistical optimality of the gradient descent algorithm of a general class of objective functions (Theorem 3.2), including PINN [12, 13] and Deep Ritz Methods [14, 2] for solving PDEs as well as Sobolev training [11, 9, 58] of kernel methods. We provide [16] a computational feasible estimator and generalize the previous statistical optimality results of gradient descent [42, 18, 19] to general Sobolev norm.

- We also characterize the acceleration effect of Sobolev loss function for learning with kernel. The acceleration happens because differential operator can enlarge the small eigenvalues for high frequency functions, leading to better condition number and faster convergence in these eigenspaces while keeping the statistical optimality. Thus when the target function have more high frequency component, the lead of PINN will become larger (Figure 3). We justify our theoretical finding with several numerical experiments (Figure 2 and Figure 4).

## 2 Problem Formulation

In this section, we formulate the problem of learning inverse problem using the kernelized gradient descent. As described previously, we aim to reconstruct a function $f^* \in \mathbb{R}^{\mathcal{X}}$ from random observations of $u^* = \mathcal{A}f^*$, where $\mathcal{A}$ is an observation process which is modeled by an operator maps from $\mathbb{R}^{\mathcal{X}}$ to $\mathbb{R}^{\mathcal{X}}$. To solve this problem, we write the operator $\mathcal{A}$ in terms of two operators $\mathcal{A}_i$ $(i = 1, 2)$ with $\mathcal{A}_1 = \mathcal{A}\mathcal{A}_2$ and build our objective function as

$$\mathbb{E}_{\mathbb{P}} \left[ \frac{1}{2} \langle u(x), \mathcal{A}_1 u(x) \rangle - \langle y, \mathcal{A}_2 u(x) \rangle \right], \tag{1}$$

where $\mathbb{P}$ is the joint distribution of $x$ and $y$ with $x$ sampled from the uniform distribution on $\mathcal{X}$ for simplicity and $y$ as the noisy observation of $f(x) = (\mathcal{A}u)(x)$. In other words, $\mathbb{E}[y|x] = f(x)$. The minimizer of objective function (1) is the ground truth function $u^* = \mathcal{A}^{-1}f$ that we are interested in.

**Learning with Kernel**  Consider the case that $u$ is parameterized by a Reproducing Kernel Hilbert Space $u_\theta(x) = \langle \theta, K_x \rangle$ (we provide standard notations of RKHS in Appendix A). At the same time, the kernel function has the following representation $K(s, t) = \sum_{i=1}^{\infty} \lambda_i e_i(s) e_j(t)$, where $e_i$ are orthogonal basis of $\mathcal{L}_2(\rho_{\mathcal{X}})$ with $\rho_{\mathcal{X}}$ being the uniform distribution over $\mathcal{X}$, where $\mathcal{L}_2$ denotes the space of all the square integrable functions. Then $e_i$ is also the eigenvector of the covariance operator $\Sigma = \mathbb{E}_{x \sim \mathbb{P}} K_x \otimes K_x$ with eigenvalue $\lambda_i > 0$, *i.e.* $\Sigma e_i = \lambda_i e_i$. Here $g \otimes h = gh^\top$ is an operator from $\mathcal{H}$ to $\mathcal{H}$ defined as $g \otimes h : f \to \langle f, h \rangle_{\mathcal{H}} g$. The covariance matrix $\Sigma$ is the core of the integral operator technique [46, 8] for kernel regression. For any $f \in \mathcal{H}$, the reproducing property gives $(\Sigma f)(z) = \langle K_z, \Sigma f \rangle_{\mathcal{H}} = \mathbb{E}[f(X)k(X, z)] = \mathbb{E}[f(X)K_x(X)]$. If we consider the mapping $S : \mathcal{H} \to L_2(dx)$ defined as a parameterization of a vast class of functions in $\mathbb{R}^{\mathcal{X}}$ via $\mathcal{H}$ through the mapping $(Sg)(x) = \langle g, K_x \rangle$ $(\Phi(x) = K_x = K(\cdot, x))$. Its adjoint operator $S^* : \mathcal{L}_2 \to \mathcal{H}$ then can be defined as $g \to \int_{\mathcal{X}} g(x) K_x \rho_X(dx)$. $\Sigma$ is the same as the self-adjoint operator $S^*S$ and the self-adjoint operator $\mathcal{L} = SS^* : \mathcal{L}_2(dx) \to \mathcal{L}_2(dx)$ can be defined as $(\mathcal{L}f)(x) = \int_{\mathcal{X}} K(x, z) f(z) \rho_{\mathcal{X}}(dz)$. Based on this notation, we present all our assumptions on the underlying kernel.

**Assumption 2.1** (Assumptions on Kernel). We assume the standard capacity condition on kernel covariance operator with a source condition about the regularity of the target function following [8]. We further assume a regularity condition for our kernel $k(\cdot, \cdot)$ via a $\ell_\infty$ embedding property follows [59, 60, 18, 21]. These conditions are stated explicitly below.

- **(a) Standard assumptions.** The kernel feature are bounded almost surely, *i.e.* $|k(x, y)| \le R$ and the observation $y$ is also bounded by $M$ almost surely.
- **(b) Capacity condition.** Consider the spectral representation of the kernel covariance operator $\sigma = \sum \lambda_i e_i \otimes e_i$, we assume polynomial decay of eigenvalues of the covariance matrix $\lambda_i \propto i^{-\alpha}$ for some $\alpha > 1$. As a result $Q = \text{tr}(\Sigma^{1/\alpha}) < \infty$.
- **(c) Source condition.** We also impose an assumption on the smoothness of the true function. There exists $\beta \in (0, 1]$ such that $u^* = \mathcal{L}^{\beta/2}\phi$ for some $\phi \in L^2$. If $u^*(x) = \langle \theta_*, K_x \rangle_{\mathcal{H}}$, the source condition can also be written as

$$\|\Sigma^{\frac{1-\beta}{2}} \theta_*\|_{\mathcal{H}} < \infty.$$

- **(d) Capacity conditions on $\mathcal{A}_i$.** For theoretical simplicity, we assume that the self-adjoint operators $\mathcal{A}_i$ are diagonalizable in the same orthonormal basis $e_i$. Thus we can assume

$$\mathcal{A}_1 = \sum_{i=1}^{\infty} p_i e_i \otimes e_i, \mathcal{A}_2 = \sum_{i=1}^{\infty} q_i e_i \otimes e_i$$

  for positive constants $p_i, q_i > 0$. We further assume $p_i \propto i^{-p}$ and $q_i \propto i^{-q}$. This commuting assumptions also made in [61, 62]. due to the Bochner's theorem. We further assume $p < 0, q < 0, \alpha + p > 0$. We refer the detailed discussion to Remark 1.
- **(e) Regularity results on RKHS.** For $\mu \in [0, 1]$, there exists $\kappa_\mu \geq 0$ such that $\Phi(x) \otimes \Phi(x) \leq k_\mu^2 R^{2\mu} \Sigma^{1-\mu}$ holds almost surely. The regularity assumption here is equivalent to $\|g\|_{L_\infty}^2 \leq \kappa_\mu^2 R^{2\mu} \|\Sigma^{1/2-\mu/2} g\|_{\mathcal{H}}^2$ and implies $\|g\|_{L_\infty} \leq \kappa_\mu R^\mu \|g\|_{\mathcal{H}}^\mu \|g\|_{L_2}^{1-\mu}$ for every $g \in \mathcal{H}$. As a consequence, we know that $\|\Sigma^{\mu/2-1/2}\Phi(x)\|_{\mathcal{H}} \leq \kappa^\mu R^\mu$ holds almost surely. [59, 21, 18]

*Remark* 1. To simplify the technical exposition, we assume that operator $\mathcal{A}_i (i = 1, 2)$ commute with the kernel covariance operator $\Sigma$. This assumption is also made in [62, 61]. Here we provide several examples that satisfy this assumption. The simplest case is $\mathcal{A}_1 = \mathcal{A}_2 = id$, which gives rise to the function regression setting. [63] assumes the operator $\mathcal{A}_i$ to be bounded operator in operator norm, which can be consider as a special case of ours. At the same time, for numerically solving a PDE/elliptic inverse problem, we take $\mathcal{A}_i$ to become the power of the Laplace operator $\Delta$, which contradicts with [63]'s assumption. If the domain is a sphere, the eigen-functions are spherical harmonics which are also the eigen-functions of a wide class of kernels, examples includes the dot product kernels [64] and the Neural Tangent Kernel [65, 66], when the data distribution is uniform distribution. When the domain is the torus, the eigen-functions are Fourier modes. If we consider a shift invariant kernel $K(x, y) = \psi(x - y)$, from Bochner's Theorem $K(x, y) = \sum_{i=1}^{n} \tilde{\psi}(w) e^{iws} e^{-iwt}$ we know that the eigen-functions are also Fourier modes. There are also works that use Green function as the kernel [67, 68], where the three operators will automatically commute with each other.

In this paper, we consider the convergence of the estimator in Sobolev norm class. We define the different Sobolev spaces via the power space approaches used in [69, 21].

**Definition 2.2** (Sobolev Norm). For $\gamma > 0$, the $\gamma$-*power space* is

$$\mathcal{H}^\gamma := \left\{ \sum_{i \geq 1} a_i \lambda_i^{\gamma/2} e_i : \sum_{i \geq 1} a_i^2 \leq \infty \right\} \subset L_2(v),$$

equipped with the $\gamma$-power norm via $\| \sum_{i \geq 1} a_i \lambda_i^{\gamma/2} e_i \|_\gamma := \left( \sum_{i \geq 1} a_i^2 \right)^{1/2}$.

It is obvious that $\|\mathcal{L}^{\gamma/2} f\|_\gamma = \|f\|_{L_2}$ and $\|f\|_\gamma \leq \|\Sigma^{\frac{1-\gamma}{2}} f\|_{\mathcal{H}}$ [21]. The source condition can also be understood as the target function $u^*$ lies in the $\beta$-power Sobolev space. The regularity condition of the kernel function implies a continuously embedding from $\mathcal{H}^\gamma \to L_\infty$. Throughout this paper, we consider the convergence rate of $\hat{u} - u^*$ in $\gamma$-power Sobolev norm ($\gamma > 0$).

## 2.1 Examples

**Sobolev Training**   [70, 9, 11] introduce the idea of training using Sobolev spaces via matching not only the function value but also the derivative of the classifier. Using different Sobolev norms as loss function has also been used widely in image processing, inverse problems, and graphics applications [71, 72, 10, 73, 74, 75]. The work of [71] discovered that different Sobolev loss functions would lead to different implicit bias and that the proper Sobolev preconditioned gradient descent can accelerate the optimization of geometry objectives [73, 74, 75]. In this paper, we discover that stochastic gradient descent over Sobolev norm loss class functions can achieve statistical optimal but proper selection of the Sobolev norm loss function can accelerate training. We call this phenomenon **Sobolev Implicit Acceleration** and discuss it in Section 4.

**Machine Learning Based PDE Solver.**   To simplify the exposition, we focus on a prototype elliptic PDE: Poisson's equation on a torus, *i.e.* $\Omega = \mathbb{T}^d = [0, 1]_{\text{per}}^d$. Our focus is on the analysis of deep-learning-based numerical methods for the elliptic equations

$$-\Delta u + u = f \quad \text{in } \Omega. \tag{2}$$

We mainly focus on analyzing Deep Ritz Method (DRM) [14] and Physics Informed Neural Network (PINN) [12, 13]. DRM solves the equation (2) via minimizing the following variational form

$$u^* = \arg\min_{u \in \mathcal{F}} \mathcal{E}^{\text{DRM}}(u) := \frac{1}{2} \int_\Omega |\nabla u|^2 + u^2 \, dx - \int_\Omega f u dx, \tag{3}$$

while PINNs solves the equation (4) via minimizing the following strong formula, *i.e* the residual of the PDE,

$$u^* = \arg\min_{u \in \mathcal{F}} \mathcal{E}^{\text{PINN}}(u) := \frac{1}{2} \int_\Omega (\Delta u - u + f)^2 \, dx, \tag{4}$$

where $u$ is minimized over a parameterized function class $\mathcal{F}$ (for example neural network). Here we consider the function class to be the RKHS space [76, 77]. [16] showed that empirical risk minimization of both objectives can achieve information theoretical optimal bounds. The objective function in 3 and 4 can be considered as special case of objective function (1). For DRM, $\mathcal{A}_1 u = \Delta u$ and $\mathcal{A}_2 u = u$ for all function $u \in \mathbb{R}^{\mathcal{X}}$. For PINN, $\mathcal{A}_1 u = \Delta^2 u$ and $\mathcal{A}_2 u = \Delta u$ for all function $u \in \mathbb{R}^{\mathcal{X}}$.

We discover that PINN convergences faster than DRM consistently due to the implicit Sobolev acceleration, matching the observation made in [78]. [61] considered semi-supervised learning using Laplacian regularization with kernel parameterization. However, this paper does not consider training with stochastic gradient descent and also does not introduce the source condition assumption that leads to different convergence rate for a hierarchical parameterization of task difficulty.

## 3 Main Theorem

We present our main results in this section, including an information theoretical lower bound and a matching upper bound with proper selected early stopping time.

### 3.1 Lower Bounds

This subsection investigates the statistical optimality of the Sobolev convergence rate of solving elliptic problem using stochastic gradient descent. We provide the information theoretical lower bound of learning the elliptic problems. Different from [15, 16], we formulate the problem in an RKHS. This leads to a different construction of hypothesis and show that [15, 16] is a special case of our lower bound using specific kernel and operator $\mathcal{A}_i (i = 1, 2)$ in Section 3.3.

**Theorem 3.1** (Lower Bound). *Let $(X, B)$ be a measurable space, $H$ be a separable RKHS on $X$ with respect to a bounded and measurable kernel $k$ and operator $\mathcal{A} = (\mathcal{A}_2^{-1} \mathcal{A}_1)$ satisfies Assumption 2.1. We have $n$ i.i.d. random observations $\{(x_i, y_i) \in \mathcal{X} \times \mathcal{Y}\}_{i=1}^n$ of $f^* = \mathcal{A}u, u \in \mathcal{H}^\gamma \cap L_\infty$, i.e. $y_i = f^*(x_i) + \eta_i$ where $\eta_i$ is a mean zero random noise satisfies the momentum assumption $\mathbb{E}|\eta|^m \leq \frac{1}{2} m! \sigma^2 L^{m-2}$ for some constants $\sigma, L > 0$. Then for all estimators $H : (\mathcal{X} \times \mathcal{Y})^{\otimes n} \to \mathcal{H}^\gamma$ satisfies*

$$\inf_H \sup_{u^*} \mathbb{E} \|H(\{(x_i, y_i)\}_{i=1}^n) - u^*\|_\gamma^2 \gtrsim n^{-\frac{(\max\{\beta,\mu\}-\gamma)\alpha}{\max\{\beta,\mu\}\alpha + 2(q-p)+1}}.$$

### 3.2 Upper Bounds

This subsection, we consider the (multiple pass) gradient descent over the empirical data of objective function (1). We aim to construct our estimator via optimizing the empirical loss function $\sum_{i=1}^n \frac{1}{2} u(x_i) \mathcal{A}_1 u(x_i) - y_i \mathcal{A}_2 u(x_i)$, where $x_i$ is sampled randomly and $y_i$ is the associated noisy observation introduced in Section 1. We consider a parameterization $u(x) = \langle u, K_x \rangle$ and $\mathcal{A}_i u(x) = \langle \mathcal{A}_i \theta, K_x \rangle_{\mathcal{H}} = \langle \theta, \mathcal{A}_i K_x \rangle_{\mathcal{H}}$ and express our empirical objective function as

$$\mathbb{E}_{\mathbb{P}_n(x,y)} \frac{1}{2} \langle u(x), \mathcal{A}_1 u(x) \rangle - \langle y, \mathcal{A}_2 u(x) \rangle = \mathbb{E}_{\mathbb{P}_n(x,y)} \frac{1}{2} \langle u, K_x \rangle \langle \mathcal{A}_1 u, K_x \rangle - y \langle \mathcal{A}_2 u, K_x \rangle$$

$$= \mathbb{E}_{\mathbb{P}_n(x,y)} \frac{1}{2} \langle u, K_x \otimes \mathcal{A}_1 K_x u \rangle - y \langle u, \mathcal{A}_2 K_x \rangle \tag{5}$$

Then the gradient descent algorithm can be written as the following procedure:

- **Initialization**: $\theta_0 = \bar{\theta}_0 = 0$, $\gamma$ is a constant to be determined later which is used as the learning rate in the algorithm.
- **Iteration**: For the $t-$th iteration, we perform the following gradient descent step

$$\theta_t = \theta_{t-1} + \gamma \frac{1}{n} \sum_{i=1}^n (y_i \mathcal{A}_2 K_{x_i} - \langle \theta_{t-1}, \mathcal{A}_1 K_{x_i} \rangle_{\mathcal{H}} K_{x_i})$$

with an averaging step $\bar{\theta}_t = (1 - \frac{1}{t}) \bar{\theta}_{t-1} + \frac{1}{t} \theta_t$.

**Remark.** Note that the optimizing dynamics considered here is not the exact gradient descent dynamics over the empirical objective. The gradient of the quadratic term $\frac{1}{n}\sum_{i=1}^{n} u(x_i)\mathcal{A}_1 u(x_i)$ should be $\frac{1}{n}\sum_{i=1}^{n}(\langle\theta_{t-1},\mathcal{A}_1 K_{x_i}\rangle_{\mathcal{H}} K_{x_i} + \langle\theta_{t-1}, K_{x_i}\rangle_{\mathcal{H}} \mathcal{A}_1 K_{x_i})$ but we take instead $\frac{1}{n}\sum_{i=1}^{n}\langle\theta_{t-1},\mathcal{A}_1 K_{x_i}\rangle_{\mathcal{H}} K_{x_i}$ in our dynamics. In the population expectation, **the two dynamics are the same** due to the commuting assumption between the kernel integral operator and operator $\mathcal{A}_1$. Without our modification, the statistical rate will become sub-optimal in some cases due to the fact that the variance in the empirical covariance matrix dominates the statistical rate. This observation matches the reason behind the sub-optimality of the Deep Ritz Method discovered in [16].

The following theorem is the main result for upper bounds with the proof details given in the appendix.

**Theorem 3.2.** *Under Assumption 2.1, we have the following three regimes shown in Figure 1.*

- *For $\beta > \frac{\alpha+2q-p-1}{\alpha}$, if we take $t = n$ and $\gamma = n^{\frac{\alpha+p}{\beta\alpha+2(p-q)+1}-1}$, we obtain the following rate*

$$\mathbb{E}[\|\bar{\theta}_t - u^*\|_\gamma^2] = O(n^{-\frac{(\beta-\gamma)\alpha}{\alpha\beta+2(p-q)+1}}).$$

- *For $\frac{\alpha+2q-p-1}{\alpha} \le \beta \le \frac{\mu\alpha+2q-p+1}{\alpha}$, if we take $t = n^{\frac{\alpha+p}{\beta\alpha+2(p-q)+1}}$ and $\gamma$ a small enough constant, we obtain the following rate*

$$\mathbb{E}[\|\bar{\theta}_t - u^*\|_\gamma^2] = O(n^{-\frac{(\beta-\gamma)\alpha}{\alpha\beta+2(p-q)+1}}).$$

- *For $\beta > \frac{\mu\alpha+2q-p+1}{\alpha}$, if we take $t = n^{\frac{\alpha+p}{\mu\alpha+p}}$ and $\gamma$ a small enough constant, we obtain the following rate*

$$\mathbb{E}[\|\bar{\theta}_t - u^*\|_\gamma^2] = O(n^{-\frac{(\beta-\gamma)\alpha}{\mu\alpha+p}}),$$

  *which is not an optimal converging rate.*

**Sketch of the Proof.** We first rewrite the averaged gradient descent in a more compact formula as $\eta_0 = 0, \eta_u = \eta_{u-1} + \gamma(\mathcal{A}_2^\top \hat{S}_n^* \hat{y} - \hat{\Sigma}_{Id,\mathcal{A}_1}\eta_{t-1})$ where $\hat{S}_n : \mathcal{H} \to \mathbb{R}^n$ is defined as $\hat{S}_n g = \frac{1}{\sqrt{n}}(g(x_1), \cdots, g(x_n))$, $\hat{\Sigma}_{\mathcal{O}_1,\mathcal{O}_2} = \frac{1}{n}\sum_{i=1}^{n}\mathcal{O}_1 K_x \otimes \mathcal{O}_2 K_x$ and $Id$ is the identity operator. For the error of GD, we consider early stopping of gradient descent algorithm as a spectral filtering [79, 18, 63, 19]. Our proof is based on standard bias-variance decomposition. For $t$ iteration, GD will behave similarly to ridge regression with $\gamma t$ regularization strength [42, 18] and this result in bias of $(\frac{1}{\gamma t})^{\frac{(\beta-\gamma)\alpha}{\alpha+p}}$. For the variance, we provide a bound which is related to the effective dimension given by $\text{tr}((\Sigma_{Id,\mathcal{A}_1} + (\frac{1}{\gamma t})I)^{-1}\Sigma_{\mathcal{A}_2^\top \mathcal{A}_2})$ and obtain a final variance of the form $\frac{1}{n}(\gamma t)^{-\frac{\gamma\alpha+p}{\alpha+p}}(\frac{1}{\gamma t})^{-\frac{1}{\alpha+p}}(\frac{1}{\gamma t})^{-\frac{p-2q}{\alpha+p}} + \frac{1}{n}(\frac{1}{\gamma t})^{-\frac{\gamma\alpha+p}{\alpha+p}}(\frac{1}{\gamma t})^{-\frac{\mu\alpha-p}{\alpha+p}}(\frac{1}{\gamma t})^{\frac{\beta\alpha-2q}{\alpha+p}}$. If we only have the first term of variance, we shall achieve information theoretical optimal bound when $t = n^{\frac{\alpha+p}{\beta\alpha+2(p-q)+1}}$. For the section term in the variance is from the convergence of empirical covariance matrix $\hat{\Sigma}_{Id,\mathcal{A}_1}$ to the population one $\Sigma_{Id,\mathcal{A}_1}$. This term can be reduced using semi-supervised learning techniques as in [80, 16].

### 3.3 Discussion and Implications of Our Theory

**Relationship with [15, 16].** [15, 16] provided a lower bound of the form $n^{-\frac{2\alpha-2s}{2\alpha-4t+d}}$ for a $2t$−th order linear PDE $\Delta^t u = f$ with solution in $H^\alpha$, evaluated in $H^s$ norm. We shall discuss the relationship between their bound with our $n^{-\frac{(\beta-\gamma)\alpha}{\beta\alpha+2(p-q)+1}}$ lower bound based on the kernel representation of Sobolev spaces. The numerator $(\beta-\gamma)$ matches the $\alpha-s$ term in [15, 16]'s lower bound and the $q-p$ term is the order of the linear PDE which matches the $t$ term in the denominator in [15, 16]'s lower bound. The spectral decay speed of kernel $\alpha$ is always relative to the dimension $d$. To understand this problem, we consider the following two examples.

For the first example, the kernel is defined on the torus $\mathbb{T}^d = [0,1]_{\text{per}}^d$. We consider the space of square integrable functions on $\mathbb{T}^d$ with mean 0 and the Matérn kernel $K_{\sigma,l,v}(x,y) = \sigma^2 \frac{2^{1-v}}{\Gamma(v)}\left(\frac{|x-y|}{l}\right)^v B_v(\frac{|x-y|}{l})$, where $B_v$ is the modified Bessel function of section kind. The covariance operator is $C_\theta = \sigma^2(-\Delta + \tau^2 I)^{-s}$ with orthonormal eigenfunctions $\phi_m(x) = e^{2\pi i\langle m,x\rangle}$ and corresponding eigenvalues $\lambda_m = \sigma^2(4\pi^2|m|^2 + \tau^2)^{-s}$ for every $m \in \mathbb{Z}^d\backslash\{0\}$ [81].

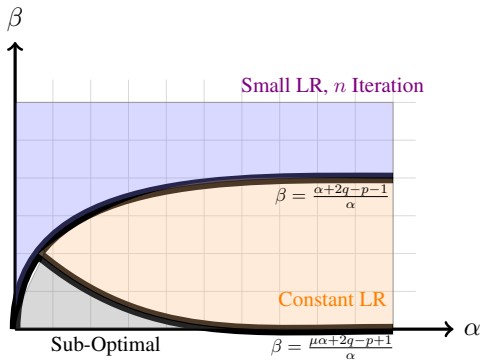

Figure 1: Phase diagram of different regimes for solving inverse problem using stochastic gradient descent. Except for the gray area, GD can achieve the information theoretic rate.

For the second example, we consider the Mercer's decomposition of a translation invariant kernel via Fourier series $K(s - t) = \frac{1}{2\pi} \sum_w \tilde{K}(w)e^{iw(s)}e^{iw(-t)}dw$. The eigenfunctions of the translation invariant kernel is the Fourier modes and the eigenvalues are the Fourier coefficients. As an example, for Neural Tangent Kernel, [82, 66, 65, 56] proved that the corresponding $\alpha = \frac{d}{d-1}$ and the eigenfunctions are spherical harmonics that diagonalize the differential equation.

For the upper bound, [16] established the convergence rate based on the *empirical process* technique [48, 59], while our paper switches to the integral operator/inverse problem technique [4, 46, 8]. An advantage of the integral operator/inverse problem technique is that it can provide convergence results with respect to a continuous scale of Sobolev norms while the empirical process technique can only be used for the Sobolev norm equivalent to the objective function.

**Relationship with [70]**  [70] also considered learning from data involving function value and gradients under the framework of least-square regularized regression in reproducing kernel Hilbert spaces. In this paper, we only have access to the noisy observation of the function values but still aim to know about the convergence rate with respect to the Sobolev norm. At the same time, we further consider an inverse problem setting with an early stopping regularization, which is not discussed in [70]. However, we introduce a commuting assumption over the differential operator with the kernel integral operator that facilitates our analysis.

**Sobolev Implicit Acceleration**  Below we discuss the implication of the choice of early stopping time $t = n^{\frac{\alpha+p}{\beta\alpha+2(p-q)+1}}$. First of all, the best early stopping time here does not depend on $\gamma$, which means the best model in different Sobolev is the same over the stochastic gradient descent path asymptotically. Secondly, all the components in an iteration step depend on the problem itself except the numerator $\alpha + p$. For differential operators, the $p$ is actually **_negative_** (differential operators have large eigenvalues over high-frequency basis). Thus we can accelerate the training via letting $p$ more negative, *i.e.* using a higher order Sobolev norm as loss can lead to earlier stopping. As an implication, the PINN achieves the statistical optimal solution faster than DRM.

**Relationship with implicit bias of frequency**  Recent work credit the success of deep learning to the fast training in low frequency components [83, 84, 85]. However, in our work, with Sobolev preconditioning, the training speed of high frequency part increases, yet achieving statistical optimality in the class of Sobolev norm. This suggests that the implicit bias of frequency is not necessary for good generalization results. We also would like refer to [86, 87] Theorem 8 for the extreme case, where the authors directly invert the population covariance matrix which leads to the same training speed in every eigen-spaces while still maintaining the statistical optimality in $\ell_2$ norm. However the preconditioning matrix in [86] is the population Fisher information matrix, which requires further sampling of unlabeled data that is not accessible in our setting.

**Discussion of the Sub-Optimal Regime**  In the sub-optimal regime, the concentration error between the empirical covariance matrix $\hat{\Sigma}_{Id,\mathcal{A}_1}$ and the population one $\Sigma_{Id,\mathcal{A}_1}$ dominates. With the observation that these concentrations have no relationship with the supervision signal, [16, 80] proposed to utilize the semi-supervised learning to reduce the error in this regime. In [16], Deep Ritz method requires semi-supervised learning while PINN does not for the exact empirical risk minimization solution. In our formulation, if $|p|$ is larger, the sub-optimal regime will become smaller, which

contradict with the observation in [16]. However [16] only considers the statistical generalization bound but doesn't take optimization into consideration. We leave designing algorithm with smaller sub-optimal regime as future work.

## 4 Sobolev implicit acceleration

The Sobolev norm has already been proposed as loss function for training neural network [9] and solving PDEs [11, 58]. However, all these papers need a further gradient information of the supervision signal. This does not fit the theoretical framework considered here and hence it is also not fair to compare their algorithms with methods without gradient supervision signal. Thus in this section, we proposed an alternative objective that can perform Sobolev training without gradient supervision loss function. The basic idea is to using an integration by parts

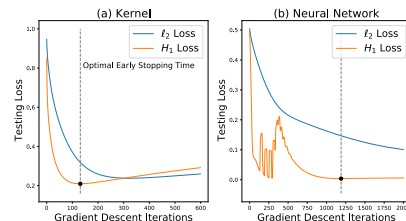

Figure 2: Sobolev Implicit Acceleration of Estimating function using kernel method and Neural Network. We observed that using Sobolev Norm as loss function can accelerate training.

$$\int |\nabla u - \nabla f|^2 dx = \int \|\nabla u\|_2^2 + 2\Delta u \cdot f + \|\nabla f\|_2^2 dx, \quad (6)$$

which leads to an objective function without the gradient of the target function. In this section, we shall show how this idea is applied to different machine learning examples.

### 4.1 Predicting a Toy Function on Torus

In this section, we conduct experiments to illustrate the Sobolev implicit acceleration for function regression. Different from the Sobolev training [9], the objective that we are interested in does not involve the gradient of the target function. As a result, we do not need to train a teacher network to provide the gradient supervision information as done in [9]. In the toy example, for simplicity we ignore the boundary terms introduced by the integral by part. Here consider estimating a function on the torus, *i.e.* a periodic function. We consider using $\int \lambda \|u - f\|^2 + \|\nabla u\|_2^2 + 2\Delta u \cdot f + \|\nabla f\|_2^2 dx$ as our objective function. The goal is to fit function $y = \sum_{i=1}^d \sin(2\pi x_i)$ using Gaussian Kernel and a simple three layer feed-forward network with tanh activation function. We randomly sampled 1000 data in 10 dimension as our dataset and run a gradient descent algorithm. Figure 2 presents our convergence result of the validation error, where the Sobolev norm have shown an acceleration effect for training.

### 4.2 Solving Partial Differential Equations

In this section, we conduct experiments to illustrate the Sobolev implicit acceleration for solving partial differential equation using PINNs [12, 58] in 3 dimensions. The example is a simple Poisson equation (static schrödinger equation) on the torus

$$\Delta u + u = f \text{ in } \mathcal{T}^d = [0, 1]_{\text{per}}^d. \quad (7)$$

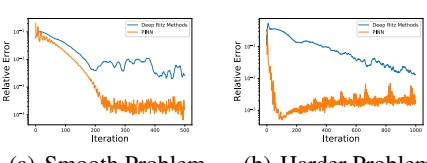

(a) Smooth Problem    (b) Harder Problem

Figure 3: We show the convergence result of PINN and Deep Ritz Method for smooth problem $\sum_{i=1}^d \sin(2\pi x)$ and harder problem $\sum_{i=1}^d \sin(4\pi x)$. PINN convergence faster than DRM for online stream input which also matches our theory and the empirical observation in [78]. The Sobolev Implicit Acceleration will becomes more significant for harder problem as our theory shows.

We first compare the Physics Informed Neural Network [12] and Deep Ritz Method [14, 2] with online random inputs. To enforce the periodic boundary conditions, we add a penalty term $\mathcal{L}_b = \int_{(x,y)\in[0,1]^2} (u(x,y,0) - u(x,y,1))^2 + (u(0,x,y) - u(1,x,y))^2 + (u(x,0,y) - u(x,1,y))^2 dxdy$ to match the periodic condition of the function value and another term $\mathcal{L}_{b,grad} = \int_{(x,y)\in[0,1]^2} (\nabla u(x,y,0) - \nabla u(x,y,1))^2 + (\nabla u(0,x,y) - \nabla u(1,x,y))^2 + (\nabla u(x,0,y) - \nabla u(x,1,y))^2 dxdy$ to match the periodic condition of the function value. We tested PINN and Deep Ritz on both $u(x) = \sum_{i=1}^d \sin(2x_i)$ and $u(x) = \sum_{i=1}^d \sin(4x_i)$. We use the same

experiment setting as [78] and keep the learning rate constantly to $1e-3$ to match our theory. 50000 data points are randomly sampled in every batch. The results are shown in Figure 3. PINN converges faster than DRM consistently in terms of iteration number and the lead seems to become significant for more oscillatory problems.

To solve equation (7), we consider minimizing the following Sobolev norm objective function

$$\mathcal{L}(u) := \lambda \|\Delta u + u - f\|_{L_2(\Omega)}^2 + \|\nabla \Delta u + \nabla u - \nabla f\|_{L_2(\Omega)}^2.$$

[11, 58] also considered using Sobolev norms as the loss function. [11] showed that the Sobolev norms exhibit an acceleration effect. However, in our setting, we cannot have random samples of $\nabla f$. To avoid information of $\nabla f$ appearing in the objective function, we perform an integration by parts that leads to the following objective function

$$\mathcal{L}_{grad} = \int \|\nabla \Delta u(x) + \nabla u(x) - \nabla f(x)\|_2^2 dx$$

$$= \int \|\nabla \Delta u(x)\|_2^2 + \|\nabla u(x)\|_2^2 + \|\nabla f(x)\|_2^2 + 2\nabla \Delta u(x)\nabla u(x) - 2\nabla u(x)\nabla f(x) - 2\nabla \Delta u(x) \cdot \nabla f(x)dx$$

$$= \int \|\nabla \Delta u(x)\|_2^2 + \|\nabla u(x)\|_2^2 + \|\nabla f(x)\|_2^2 + 2\nabla \Delta u(x)\nabla u(x) + 2\Delta u(x)f(x) + 2\Delta \Delta u(x) \cdot f(x)dx.$$

We conduct the Sobolev training with the objective function $\mathcal{L}_{pinn} + \lambda \mathcal{L}_{grad} + \lambda_1 \mathcal{L}_b + \lambda \mathcal{L}_{b,grad}$ and compare it with PINN and DRM. Following mostly the experiment setting in [78], we fix 3000 random samples as the dataset and run stochastic gradient descent with batchsize 50. The result presented in Figure 4 show the Sobolev implicit acceleration, i.e., the gradient dynamic of higher order Sobolev norm convergence faster. We do not scale the Sobolev training to online setting as under large batch size the Sobolev training consume too much memory at this point.

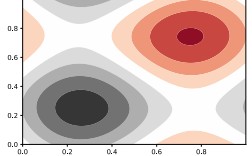

(a) Solution by Sobolev Training.

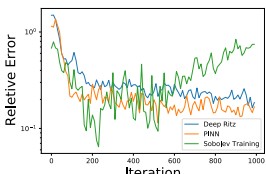

(b) Convergence Speed.

Figure 4: Solving equation (7) in 3 dimension with 3000 fixed samples using Deep Ritz Method [14], Physics-Informed Neural Network [12] and Sobolev Training.

## 5 Conclusion and Discussion

In this paper, we consider the statistical optimality of gradient descent for solving elliptic inverse problem using a general class of objective functions. Although we can achieve statistical optimality of gradient descent using all the objective functions with proper early stopping time, the early stopping iteration strategy for the optimal solution behaves differently as a function of the sample size. For instance, we observed that PINN convergences faster than the DRM method. Generally speaking, by using a higher order Sobolev norm as loss function, one can accelerate training. The reason is that the differential operator can counteract the kernel integral operator, leading to better condition number for optimization. We call this phenomena *Sobolev implicit acceleration*.

Although we have shown the Sobolev implicit acceleration on several simple examples, the $\Delta^s u$ term is hard to compute in high dimensions, scalable Sobolev training without gradient supervision in higher dimension remains as future work. However, we believe that this direction is promising. For example, we can use MIM method [88, 89] to accelerate the training. It is also interesting to generalize our results beyond GD, for example to mirror descent [90] and accelerated gradient descent [91]. In this paper, we did not consider operators with continuous spectrum and it will be interesting to extend our results using the techniques in [92]. Due to technical issue, we have not considered the batch stochastic gradient descent. It will be interesting to characterize the condition under which the stochastic noise in gradient does not degrade the optimal bounds that we obtain. At the same time, we also want to investigate more complex nonlinear inverse problems as [93, 94] considered. It is also interesting to consider inverse problem arising from integral equation where $p > 0$.

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
