# Appendix.

The appendix is constructed as follows:

- In Appendix A, we introduce the basic notations of Reproducing Kernel Hilbert space and the associated kernel integral operator. We also put a discussion of how differential operators and Sobolev spaces relates to the kernel setting we considered as a preliminary.
- In Appendix B.1, we consider the statistical optimality of the early stopped gradient descent algorithm. We bound the difference of the gradient descent.
- In Appendix C, we provide our proof for the lower bound in Section 3.1 using the Fano method.

## A Preliminaries and Notations

This section starts with an overview of reproducing kernel Hilbert space, including Mercer's decomposition, the integral operator techniques[46, 4, 8, 21] and the relationship between RKHS and the Sobolev space[95]. In order to fit the objective function we considered, we did a slight modification to the original integral operator technique[46, 4, 8].

### A.1 Reproducing Kernel Hilbert Space

We consider a Hilbert space $\mathcal{H}$ with inner product $\langle, \rangle_{\mathcal{H}}$ is a separable Hilbert space of functions $\mathcal{H} \subset \mathbb{R}^{\mathcal{X}}$. We call this space a Reproducing Kernel Hilbert space if $f(x) = \langle f, K_x \rangle_{\mathcal{H}}$ for all $K_x \in \mathcal{H} : t \to K(x,t), x \in \mathcal{X}$. Now we consider a distribution $\rho$ on $\mathcal{X} \times \mathcal{Y}(\mathcal{Y} \subset \mathbb{R})$ and denote $\rho_X$ as the margin distribution of $\rho$ on $\mathcal{X}$. We further assume $\mathbb{E}[K(x,x)] < \infty$ and $\mathbb{E}[Y^2] < \infty$. We define $g \otimes h = gh^\top$ is an operator from $\mathcal{H}$ to $\mathcal{H}$ defined as

$$g \otimes h : f \to \langle f, h \rangle_{\mathcal{H}} g.$$

At the same time, we knows that

$$\|f \otimes g\| = \|f\|_{\mathcal{H}} \|g\|_{\mathcal{H}}$$

holds for all $f, g \in \mathcal{H}$.

The integral operator technique[46, 8] consider the covariance operator on the Hilbert space $\mathcal{H}$ defined as $\Sigma = \mathbb{E}_{\rho_X} K_x \otimes K_x$. Then for all $f \in \mathcal{H}$, using the reproducing property, we know that

$$(\Sigma f)(z) = \langle K_z, \Sigma f \rangle_{\mathcal{H}} = \mathbb{E}[f(X)k(X,z)] = \mathbb{E}[f(X)K_x(X)].$$

If we consider the mapping $S : \mathcal{H} \to L_2(dx)$ defined as a parameterization of a vast class of functions in $\mathbb{R}^{\mathcal{X}}$ via $\mathcal{H}$ through the mapping $(Sg)(x) = \langle g, K_x \rangle$ $(\Phi(x) = K_x = K(\cdot, x))$. Its adjoint operator $S^*$ then can be defined as $S^* : \mathcal{L}_2 \to \mathcal{H} : g \to \int_{\mathcal{X}} g(x) K_x \rho_X(dx)$ and at the same time $\Sigma$ is the same as the self-adjoint operator $S^*S$ and the self-adjoint operator $\mathcal{L} = SS^* : L_2 \to L_2$ can be defined as

$$(\mathcal{L}f)(x) = (SS^*f)(x) = \int_{\mathcal{X}} K(x,z)f(z)d\rho_X(x), \forall f \in L_2$$

Next we consider the eigen-decomposition of the integral operator $\mathcal{L}$ via Mecer's Theorem. There exists an orthonormal basis $\{\psi_i\}$ of $\mathcal{L}_2(\mathcal{X})$ consisting of eigenfunctions of kernel integral operator $\mathcal{L}$. At the same time, the kernel function have the following representation $K(s,t) = \sum_{i=1}^{\infty} \lambda_i e_i(s) e_j(t)$ where $e_i$ are orthogonal basis of $\mathcal{L}_2(\rho_X)$. Then $e_i$ is also the eigenvector of the covariance operator $\Sigma$ with eigenvalue $\lambda_i > 0$, *i.e.* $\Sigma e_i = \lambda_i e_i$.

## B Proof of the Upper Bound

In this section, we consider the convergence of the gradient descent algorithm to the target function 1. In particular, we consider the gradient descent as a special case of a wider class of spectral filter algorithms [49, 79, 18, 19]. In our inverse problem setting, the spectral filter is defined as the estimator of the following form for $\lambda > 0$,

$$\hat{q}_\lambda = g_\lambda(\hat{\Sigma}_{Id, \mathcal{A}_1}) \mathcal{A}_2 \hat{S}_n^* \hat{y},$$

where $\hat{S}_n g = (g(x_1), \cdot, g(x_n))$ (leads to $\hat{S}_n^*$ maps from $\mathbb{R}^n$ to $\mathcal{H}$ via $\hat{S}_n^*(a_1, a_2, \cdots, a_n) = \frac{1}{n} \sum_{i=1}^{n} a_n K_{x_n}$), $\hat{\Sigma}_{\mathcal{O}_1, \mathcal{O}_2} = \frac{1}{n} \sum_{i=1}^{n} \mathcal{O}_1 K_x \otimes \mathcal{O}_2 K_x$ and $Id$ is the identity operator. The function $q_\lambda : \mathbb{R}^+ \to \mathbb{R}^+$ is a function known as *filter*, which is an approximation of $x^{-1}$ controlled by

$\lambda$. We further define the error of approximation via $r_\lambda(x) = 1 - xq_\lambda(x)$. **Spectral Filters** need the function $q_\lambda$ further satisfies

$$\lambda q_\lambda(x) \le c_q, r_\lambda(x)x^u \le c_q\lambda^u, \forall x > 0, \lambda > 0, u \in [0,1],$$

for some positive $c_q > 0$. Next we show that the averaged gradient descent can be considered as spectral filter algorithm with filter $q^\eta(x) = \left(1 - \frac{1-(1-\gamma x)^t}{\gamma t x}\right)\frac{1}{x}$. Let us consider the gradient descent $\eta_0 = 0, \eta_u = \eta_{u-1} + \gamma(\mathcal{A}_2^\top \hat{S}_n^* \hat{y} - \hat{\Sigma}_{Id,\mathcal{A}_1}\eta_{t-1})$, then

$$
\eta_t = (I - \gamma\hat{\Sigma}_{Id,\mathcal{A}_1})\eta_{t-1} + \gamma\mathcal{A}_2^\top \hat{S}_n^* \hat{y} = \gamma\sum_{k=0}^{t-1}(I - \gamma\hat{\Sigma}_{Id,\mathcal{A}_1}))^k\mathcal{A}_2^\top \hat{S}_n^* \hat{y} \tag{8}
$$
$$
= \left[I - (I - \gamma\hat{\Sigma}_{Id,\mathcal{A}_1}))^t\right](\hat{\Sigma}_{Id,\mathcal{A}_1}))^{-1}\mathcal{A}_2^\top \hat{S}_n^* \hat{y}
$$

and

$$
\bar{\eta}_t = \frac{1}{t}\sum_{i=0}^{t}\eta_i = \frac{1}{t}\sum_{i=0}^{t}\left[I - (I - \gamma\hat{\Sigma}_{Id,\mathcal{A}_1}))^i\right](\hat{\Sigma}_{Id,\mathcal{A}_1}))^{-1}\mathcal{A}_2^\top \hat{S}_n^* \hat{y}. \tag{9}
$$

Thus if we take the filter $q_t(x) = \frac{1}{x}\left(1 - \frac{1-(1-\gamma x)^t}{\gamma t x}\right)$, we can have $\bar{\eta} = q_t(\Sigma_{Id,\mathcal{A}_1})\mathcal{A}_2^\top \hat{S}_n^* \hat{y}$. At the same time $x^u r_t(x) = x^u(1 - xq_t(x)) = x^u\frac{1-(1-\gamma t)^t}{\gamma t x} \le \frac{(\gamma t x)^{1-u}}{\gamma t x}x^u = \frac{1}{(\gamma t)^u}$. Thus we can consider the gradient descent algorithm for the inverse problem as a spectral filtering algorithm.

Next, we compare the spectral filter of early stopped gradient descent with ridge regression and decompose the risk to bias and variance terms. Via bounding the bias and variance separately, we can achieve information theoretical optimal upper bound for such problems.

## B.1 Convergence Of the Gradient Descent Algorithm

To conduct our proof of the upper bound, we consider $g_\lambda = (\Sigma_{Id,\mathcal{A}_1} + \lambda I)^{-1}\mathcal{A}_2 S^* f_\rho$ and decompose the error as $\underbrace{(g_\lambda - u^*)}_{\text{Bias}} + \underbrace{(\hat{q}_\lambda - g_\lambda)}_{\text{Variance}}$. We first bound the bias in the general Sobolev norm then come to bound the variance.

### B.1.1 Auxiliary Lemmas

We first introduce several auxiliary lemmas which aims to bound different quantities according to the effective dimension/capacity of the kernel covariance operator. We define $\mathcal{N}(\lambda) = \mathbb{E}_x\|(\Sigma_{\mathcal{A}_1} + \lambda)^{-1/2}\mathcal{A}_2 K_x\|_H^2 = \text{Tr}((\Sigma_{\mathcal{A}_1} + \lambda)^{-1}\Sigma_{\mathcal{A}_2\mathcal{A}_2})$, $\mathcal{N}_\infty^1(\lambda) = \sup_{x\in\rho(x)}\|(\Sigma_{Id,\mathcal{A}_1} + \lambda)^{-1/2}\mathcal{A}_2 K_x\|_H^2$ and $\mathcal{N}_\infty^2(\lambda) = \sup_{x\in\rho(x)}\|(\Sigma_{Id,\mathcal{A}_1} + \lambda)^{-1/2}K_x\|_H^2$ which are important important quantities used to bound the variance of our estimator.

**Lemma B.1.** *There exists a constant $D$ such that the following inequality is satisfied for $\lambda > 0$,*

$$\mathcal{N}(\lambda) \le D(\lambda)^{-\frac{1}{p+\alpha} + \frac{p-2q}{p+\alpha}}$$

*Proof.* We use the spectral representation to bound the effective dimension $\mathcal{N}(\lambda)$ as

$$
\mathcal{N}(\lambda) = \text{Tr}((\Sigma_{\mathcal{A}_1} + \lambda)^{-1}\Sigma_{\mathcal{A}_2\mathcal{A}_2}) = \sum_{i=1}^{\infty}\frac{\mu_i q_i^2}{\mu_i p_i + \lambda}
$$
$$
\lesssim \sum_{i=1}^{\infty}\frac{i^{-\alpha-2q}}{i^{-\alpha-p} + \lambda} \le \int_0^\infty \frac{\tau^{p-2q}}{1 + \lambda(\tau^{\alpha+p})}d\tau \tag{10}
$$
$$
= (\lambda)^{-\frac{1}{p+\alpha}}\int_0^\infty \frac{(\lambda)^{\frac{2q-p}{p+\alpha}}\tau^{p-q}}{1 + \tau^{\alpha+p}} = \Omega\left((\lambda)^{-\frac{1}{p+\alpha} - \frac{p-2q}{p+\alpha}}\right)
$$

$\square$

**Lemma B.2.** *There exists a constant $D$ such that the following inequality is satisfied for $\lambda > 0$,*

$$Trace((\Sigma_{\mathcal{A}_1} + \lambda)^{-1}\Sigma_{Id,Id}) \le D\lambda^{\frac{-p-1}{p+\alpha}}$$

*Proof.* Similarly we use the spectral representation to bound the LHS as

$$\text{Trace}((\Sigma_{\mathcal{A}_1} + \lambda)^{-1}\Sigma_{Id,Id}) = \sum_{i=1}^{\infty} \frac{\mu_i}{\mu_i p_i + \lambda}$$

$$\lesssim \sum_{i=1}^{\infty} \frac{i^{-\alpha}}{i^{-\alpha-p} + \lambda} \leq \int_0^{\infty} \frac{\tau^p}{1 + \lambda(\tau^{\alpha+p})} d\tau \qquad (11)$$

$$= (\lambda)^{-\frac{1}{p+\alpha}} \int_0^{\infty} \frac{(\lambda)^{\frac{-p}{p+\alpha}}\tau^p}{1 + \tau^{\alpha+p}} = \Omega\left(\lambda^{\frac{-p-1}{p+\alpha}}\right)$$

$\square$

**Lemma B.3.** *We denote the following quantity by $\mathcal{N}_{\infty}^1$, $\mathcal{N}_{\infty}^2$ and $\mathcal{N}_{\infty}^3$ can be bounded by*

- $\mathcal{N}_{\infty}^1(\lambda) = \sup_{x \in \rho(x)} \|(\Sigma_{Id,\mathcal{A}_1} + \lambda)^{-1/2} K_x\|_H^2 \leq \|k_v^{\alpha}\|_{\infty}^2 \lambda^{-\frac{\mu\alpha+p}{\alpha+p}}$,
- $\mathcal{N}_{\infty}^2(\lambda) = \sup_{x \in \rho(x)} \|(\Sigma_{Id,\mathcal{A}_1} + \lambda)^{-1/2} \mathcal{A}_2 K_x\|_H^2 \leq \|k_v^{\alpha}\|_{\infty}^2 \lambda^{-\frac{\mu\alpha+p+2q}{\alpha+p}}$,
- $\mathcal{N}_{\infty}^3(\lambda) = \sup_{x \in \rho(x)} \|(\Sigma_{Id,\mathcal{A}_1} + \lambda)^{-1/2} \mathcal{A}_1 K_x\|_H^2 \leq \|k_v^{\alpha}\|_{\infty}^2 \lambda^{-\frac{\mu\alpha+3p}{\alpha+p}}$.

*Proof.* We can also prove the bound from the spectral formulation and the $l_{\infty}$ embedding property of the kernel function

$$\|(\Sigma_{Id,\mathcal{A}_1} + \lambda)^{-1/2} K_x\|_H^2 = \sum_{i \geq 1} \frac{\mu_i}{\mu_i p_i + \lambda} e_i^2(x)$$

$$\leq \left(\sum_{i \geq 1} \mu_i^{\mu} e_i^2(x)\right) \sup_{i \geq 1} \frac{\mu_i^{1-\mu}}{\mu_i p_i + \lambda} \lesssim \left(\sum_{i \geq 1} \mu_i^{\mu} e_i^2(x)\right) \sup_{i \geq 1} \frac{i^{-(1-\mu)\alpha}}{i^{-\alpha-p} + \lambda}$$

$$\leq \lambda^{-\frac{\mu\alpha+p}{\alpha+p}} \|k_v^{\mu}\|_{\infty}^2, \qquad (12)$$

and

$$\|(\Sigma_{Id,\mathcal{A}_1} + \lambda)^{-1/2} \mathcal{A}_2 K_x\|_H^2 = \sum_{i \geq 1} \frac{\mu_i q_i^2}{\mu_i p_i + \lambda} e_i^2(x)$$

$$\leq \left(\sum_{i \geq 1} \mu_i^{\mu} e_i^2(x)\right) \sup_{i \geq 1} \frac{\mu_i^{1-\mu} q_i^2}{\mu_i p_i + \lambda} \lesssim \left(\sum_{i \geq 1} \mu_i^{\mu} e_i^2(x)\right) \sup_{i \geq 1} \frac{i^{-(1-\mu)\alpha-2q}}{i^{-\alpha-p} + \lambda}$$

$$\leq \lambda^{-\frac{\mu\alpha+p+2q}{\alpha+p}} \|k_v^{\mu}\|_{\infty}^2. \qquad (13)$$

Similarly we have

$$\|(\Sigma_{Id,\mathcal{A}_1} + \lambda)^{-1/2} \mathcal{A}_1 K_x\|_H^2 = \sum_{i \geq 1} \frac{\mu_i p_i^2}{\mu_i p_i + \lambda} e_i^2(x)$$

$$\leq \left(\sum_{i \geq 1} \mu_i^{\mu} e_i^2(x)\right) \sup_{i \geq 1} \frac{\mu_i^{1-\mu} p_i^2}{\mu_i p_i + \lambda} \lesssim \left(\sum_{i \geq 1} \mu_i^{\mu} e_i^2(x)\right) \sup_{i \geq 1} \frac{i^{-(1-\mu)\alpha-2p}}{i^{-\alpha-p} + \lambda}$$

$$\leq \lambda^{-\frac{\mu\alpha+2p}{\alpha+p}} \|k_v^{\mu}\|_{\infty}^2. \qquad (14)$$

$\square$

**Lemma B.4.** *For all $\lambda > 0$, we have*

$$\|\Sigma^{\frac{1-\gamma}{2}}(\Sigma_{Id,\mathcal{A}_1} + \lambda)^{-1/2}\|^2 \leq \lambda^{-\frac{\gamma\alpha+p}{\alpha+p}}$$

*Proof.* We first bound $\|\Sigma^{\frac{1-\gamma}{2}}(\Sigma_{Id,\mathcal{A}_1}+\lambda)^{-1/2}\|^2$

$$\|\Sigma^{\frac{1-\gamma}{2}}(\Sigma_{Id,\mathcal{A}_1}+\lambda)^{-1/2}\|^2 = \sup_{i\geq 1}\frac{\mu_i^{1-\gamma}}{\mu_i p_i+\lambda} \lesssim \sup_{i\geq 1}\frac{i^{-(1-\gamma)\alpha}}{i^{-\alpha-p}+\lambda} \leq \lambda^{-\frac{\gamma\alpha+p}{\alpha+p}}$$

$\square$

**Lemma B.5.** *With probability $1-e^{-\tau}$, we have*

$$\|(\Sigma_{Id,\mathcal{A}_1}+\lambda)^{-1/2}(\hat{\Sigma}_{Id,\mathcal{A}_1}-\Sigma_{Id,\mathcal{A}_1})(\Sigma_{Id,\mathcal{A}_1}+\lambda)^{-1/2}\|^2 \lesssim \sqrt{\frac{\tau\sqrt{\mathcal{N}_\infty^1(\lambda)\mathcal{N}_\infty^3(\lambda)}}{n}}$$

*and as a consequence once $n \gtrsim \tau\lambda^{-\frac{\mu\alpha+2p}{\alpha+p}}$, we'll have*

$$\frac{1}{2} \leq \|(\Sigma_{Id,\mathcal{A}_1}+\lambda)^{1/2}(\hat{\Sigma}_{Id,\mathcal{A}_1}+\lambda)^{-1/2}\| \leq 2, \frac{1}{2} \leq \|(\Sigma_{Id,\mathcal{A}_1}+\lambda)^{-1/2}(\hat{\Sigma}_{Id,\mathcal{A}_1}+\lambda)^{1/2}\| \leq 2.$$

*Proof.* We utilize the concentration result for Hilbert space valued random variable [96] to prove the bound here. Now, we consider the operator $C_x : \mathcal{H} \to \mathcal{H}$ the operator defined by

$$C_x f := \mathcal{A}_1 f(x)k(x,\cdot) = \langle f, \mathcal{A}_1 K_x\rangle K_x,$$

and consider the random variable $\xi_x := (\Sigma_{Id,\mathcal{A}_1}+\lambda)^{1/2}C_x(\Sigma_{Id,\mathcal{A}_1}+\lambda)^{-1/2}$. From definition, we know that

$$\begin{aligned}
\xi_x f &= (\Sigma_{Id,\mathcal{A}_1}+\lambda)^{1/2}C_x(\Sigma_{Id,\mathcal{A}_1}+\lambda)^{-1/2}f\\
&= \left\langle f, (\Sigma_{Id,\mathcal{A}_1}+\lambda)^{-1/2}\mathcal{A}_1 K_x\right\rangle(\Sigma_{Id,\mathcal{A}_1}+\lambda)^{1/2}K_x \qquad (15)\\
&= ((\Sigma_{Id,\mathcal{A}_1}+\lambda)^{1/2}K_x \otimes (\Sigma_{Id,\mathcal{A}_1}+\lambda)^{-1/2}\mathcal{A}_1 K_x)f.
\end{aligned}$$

At the same time, we know that $\|f\otimes g\| = \|f\|_\mathcal{H}\|g\|_\mathcal{H}$ for all $f,g \in \mathcal{H}$, thus utilizing the concentration results for Hilbert space valued random variable, we have

$$\|(\Sigma_{Id,\mathcal{A}_1}+\lambda)^{-1/2}(\hat{\Sigma}_{Id,\mathcal{A}_1}-\Sigma_{Id,\mathcal{A}_1})(\Sigma_{Id,\mathcal{A}_1}+\lambda)^{-1/2}\|^2 \lesssim \sqrt{\frac{\tau\sqrt{\mathcal{N}_\infty^1(\lambda)\mathcal{N}_\infty^3(\lambda)}}{n}}.$$

From Lemma B.3, we know that $\mathcal{N}_\infty^1(\lambda) = \sup_{x\in\rho(x)}\|(\Sigma_{Id,\mathcal{A}_1}+\lambda)^{-1/2}K_x\|_H^2 \leq \|k_v^\alpha\|_\infty^2\lambda^{-\frac{\mu\alpha+p}{\alpha+p}}$, and $\mathcal{N}_\infty^3(\lambda) = \sup_{x\in\rho(x)}\|(\Sigma_{Id,\mathcal{A}_1}+\lambda)^{-1/2}\mathcal{A}_1 K_x\|_H^2 \leq \|k_v^\alpha\|_\infty^2\lambda^{-\frac{\mu\alpha+3p}{\alpha+p}}$. Thus once $n \gtrsim \tau\lambda^{-\frac{\mu\alpha+2p}{\alpha+p}}$, we'll have

$$\frac{1}{2} \leq \|(\Sigma_{Id,\mathcal{A}_1}+\lambda)^{1/2}(\hat{\Sigma}_{Id,\mathcal{A}_1}+\lambda)^{-1/2}\| \leq 2, \frac{1}{2} \leq \|(\Sigma_{Id,\mathcal{A}_1}+\lambda)^{-1/2}(\hat{\Sigma}_{Id,\mathcal{A}_1}+\lambda)^{1/2}\| \leq 2.$$

$\square$

**Theorem B.6** (Bernstein's Inequality). *Let $(\Omega, \mathcal{B}, P)$ be a probability space, $H$ be a separable Hilbert space, and $\xi : \Omega \to H$ with*

$$\mathbb{E}_P\|\xi\|_H^m \leq \frac{1}{2}m!\sigma^2 L^{m-2}$$

*for all $m \geq 2$. Then, for $\tau \geq 1$ and $n \geq 1$, the following concentration inequality is satisfied*

$$\mathcal{P}^n\left[\left\|\frac{1}{n}\sum_{i=1}^n \xi(\omega_i) - \mathbb{E}_P\xi\right\|_H^2 \geq 32\frac{\tau^2}{n}\left(\sigma^2+\frac{L^2}{n}\right)\right] \leq 2e^{-\tau}$$

**Lemma B.7** (Lemma 25 in [21]). *For $\lambda > 0$ and $0 \leq \alpha \leq 1$, the function $f_{\lambda,\alpha} : [0,\infty) \to \mathbb{R}$ be defined by $f_{\lambda,\alpha}(t) := \frac{t^\alpha}{\lambda+t}$. In the case $\alpha = 0$ the function is decreasing and for $\alpha = 1$ the function is increasing. Furthermore*

$$\lambda^{\alpha-1}/2 \leq \sup_{t\geq 0}f_{\lambda,\alpha}(t) \leq \lambda^{\alpha-1}$$

*for $0 < \alpha < 1$ the function attain its supremum at $t^* = \frac{\lambda\alpha}{1-\alpha}$*

*Proof.* For completeness, we provide the proof here. For function $f_{\lambda,\alpha}(t) := \frac{t^\alpha}{\lambda+t}$ with $0 < \alpha < 1$, we know the derivative of it is $f'_{\lambda,\alpha}(t) = \frac{\alpha t^{\alpha-1}(\alpha+t)-t^\alpha}{(\lambda+t)^2}$. The derivative $f'_{\lambda,\alpha}$ has a unique root at $t^* = \alpha\lambda/(1-\alpha)$. $f_{\lambda,\alpha}$ attains global maximum at $t^*$ and

$$\sup_{t\geq 0} f_{\lambda,\alpha}(t) = f_{\lambda,\alpha}(t^*) = \lambda^{\alpha-1}\alpha^\alpha(1-\alpha)^{1-\alpha} \leq \lambda^{\alpha-1}.$$

At the same time, $(\alpha^\alpha(1-\alpha)^{1-\alpha})' = \alpha^\alpha(1-\alpha)^{1-\alpha}\log\left(\frac{\alpha}{1-\alpha}\right)$ thus $\alpha^\alpha(1-\alpha)^{1-\alpha}$ achieves minimum $\frac{1}{2}$ when $\alpha = \frac{1}{2}$. Thus we know $\lambda^{\alpha-1}/2 \leq \sup_{t\geq 0} f_{\lambda,\alpha}(t)$. $\square$

### B.1.2 Bias

In this section, we consider the bias introduced by the regularization factor, *i.e.* the difference between $g_\lambda = (\Sigma_{Id,\mathcal{A}_1} + \lambda I)^{-1}\mathcal{A}_2 S^* f_\rho$ and the ground truth solution $\mathcal{A}_1^{-1}\mathcal{A}_2 f_\rho$.

**Lemma B.8.** *If $u^* = \mathcal{A}_1^{-1}\mathcal{A}_2 f_\rho \in [H]^\beta$ holds, then for all $0 \leq \gamma \leq \beta$ and $\lambda > 0$, the following bounds holds*

$$\|g_\lambda - \mathcal{A}_1^{-1}\mathcal{A}_2 f_\rho\|_\gamma \lesssim \lambda^{\frac{(\beta-\gamma)\alpha}{\alpha+p}}\|u^*\|_{[H]^\beta}.$$

*Here $g_\lambda = (\Sigma_{Id,\mathcal{A}_1} + \lambda I)^{-1}\mathcal{A}_2 S^* f_\rho$.*

*Proof.* Since $u^* = \mathcal{A}_1^{-1}\mathcal{A}_2 f_\rho \in [H]^\beta$, we can use the spectral representation $u^* = \sum_{i=1}^n a_i e_i$ with $\|u^*\|_{[H]^\beta} = \sum_{i=1}^\infty \mu_i^{-\beta}a_i$. At the same time $\mathcal{A}_2 f_\rho = \mathcal{A}_1 u^* = \sum_{i=1}^n a_i p_i e_i$. We also observe that the matrix $(\Sigma_{Id,\mathcal{A}_1} + \lambda I)^{-1}$ have the spectral representation $(\Sigma_{Id,\mathcal{A}_1} + \lambda I)^{-1} = \sum_{i=1}^\infty (\mu_i p_i + \lambda)^{-1}e_i \otimes e_i$ and leads to the spectral representation of the solution

$$g_\lambda = (\Sigma_{Id,\mathcal{A}_1} + \lambda I)^{-1}\mathcal{A}_2 S^* f_\rho = \sum_{i=1}^\infty \frac{\mu_i q_i}{\mu_i p_i + \lambda}\frac{p_i}{q_i}a_i e_i = \sum_{i=1}^\infty \frac{\mu_i p_i}{\mu_i p_i + \lambda}a_i e_i$$

Then we can bound the bias via the spectral representation

$$\begin{aligned}
\|g_\lambda - \mathcal{A}_1^{-1}\mathcal{A}_2 f_\rho\|_\gamma^2 &= \|(\Sigma_{Id,\mathcal{A}_1} + \lambda I)^{-1}\mathcal{A}_2 S^* f_\rho - \mathcal{A}_1^{-1}\mathcal{A}_2 f_\rho\|_\gamma^2 \\
&= \left\|\sum_{i=1}^\infty \frac{\mu_i p_i}{\mu_i p_i + \lambda}a_i e_i - a_i e_i\right\|^2 = \left\|\sum_{i=1}^\infty \frac{\lambda}{\mu_i p_i + \lambda}a_i e_i\right\|_\gamma^2 \\
&= \sum_{i=1}^\infty \left(\frac{\lambda}{\mu_i p_i + \lambda}a_i\right)^2 \mu_i^{-\gamma} \\
&= \lambda^2 \left(\sup_{i\geq 1} \frac{i^{-\alpha(\frac{\beta-\gamma}{2})}}{\lambda + i^{-\alpha-p}}\right)^2 \sum_{i\geq 1}\mu_i^{-\beta}a_i^2 \leq \lambda^{\frac{(\beta-\gamma)\alpha}{\alpha+p}}\|u^*\|_{[H]^\beta}^2
\end{aligned} \tag{16}$$

$\square$

In this section, we also bound a bias over the energy function $\|\mathcal{A}_1 g_\lambda - \mathcal{A}_2 f_\rho\|_2^2$, which will be used in bounding the variance term.

**Lemma B.9.** *If $u^* = \mathcal{A}_1^{-1}\mathcal{A}_2 f_\rho \in [H]^\beta$ holds, then for all $0 \leq \gamma \leq \beta$ and $\lambda > 0$, the following bounds holds*

$$\|\mathcal{A}_1 g_\lambda - \mathcal{A}_2 f_\rho\|_2 \lesssim \lambda^{\frac{\beta\alpha-2p}{2(\alpha+p)}}\|u^*\|_{[H]^\beta}.$$

*Here $g_\lambda = (\Sigma_{Id,\mathcal{A}_1} + \lambda I)^{-1}\mathcal{A}_2 S^* f_\rho$.*

*Proof.* As discussed in the proof of Lemma B.8, we have the spectral representation of $g_\lambda$ as

$$g_\lambda = (\Sigma_{Id,\mathcal{A}_1} + \lambda I)^{-1}\mathcal{A}_2 S^* f_\rho = \sum_{i=1}^\infty \frac{\mu_i q_i}{\mu_i p_i + \lambda}\frac{p_i}{q_i}a_i e_i = \sum_{i=1}^\infty \frac{\mu_i p_i}{\mu_i p_i + \lambda}a_i e_i$$

Thus $\mathcal{A}_1 g_\lambda - \mathcal{A}_2 f_\rho = \sum_{i=1}^\infty \left(\frac{\mu_i p_i^2}{\mu_i p_i + \lambda} - p_i\right)a_i e_i = -\sum_{i=1}^\infty \left(\frac{p_i\lambda}{\mu_i p_i + \lambda}\right)a_i e_i$ and we can have the bound of the bias in the energy norm as

$$\|\mathcal{A}_1 g_\lambda - \mathcal{A}_2 f_\rho\|_2^2 = \left\| \sum_{i=1}^\infty \left( \frac{p_i \lambda}{\mu_i p_i + \lambda} - p_i \right) a_i e_i \right\|_2^2$$

$$= \sum_{i=1}^\infty \left( \frac{p_i \lambda}{\mu_i p_i + \lambda} a_i \right)^2 = \lambda^2 \left( \sup_{i \geq 1} \frac{i^{-(\frac{\alpha\beta}{2})-p}}{\lambda + i^{-\alpha-p}} \right)^2 \sum_{i \geq 1} \mu_i^{-\beta} a_i^2$$

$$\lesssim \lambda^{\frac{\beta\alpha - 2p}{\alpha+p}} \|u^*\|_{[H]^\beta}.$$

$\square$

### B.1.3 Variance

In this section, we bound the variance which defined as the difference between between $g_\lambda = (\Sigma_{Id,\mathcal{A}_1} + \lambda I)^{-1} \mathcal{A}_2 S^* f_\rho$ and $\hat{g}_\lambda = q_\lambda(\hat{\Sigma}_{Id,\mathcal{A}_1}) \mathcal{A}_2 \hat{S}^* y$ at the scale $O \left( \frac{(\sigma^2 + R^2 \lambda^{2r}) \mathcal{N}(\lambda_q)}{n} + \lambda^{\frac{(\beta-\gamma)\alpha - \mu\alpha - p}{\alpha+p}}}{n} + o(\frac{1}{n}) \right)$. We first did the following decomposition

$$\Sigma^{\frac{1-\gamma}{2}} (g_\lambda - \hat{g}_\lambda) = \Sigma^{\frac{1-\gamma}{2}} q_\lambda(\hat{\Sigma}_{Id,\mathcal{A}_1})(\mathcal{A}_2 \hat{S}^* y - (\hat{\Sigma}_{Id,\mathcal{A}_1}) g_\lambda) + \Sigma^{\frac{1-\gamma}{2}} \left[ g_\lambda(\hat{\Sigma}_{Id,\mathcal{A}_1}) \hat{\Sigma}_{Id,\mathcal{A}_1} - I \right] g_\lambda$$

$$= \Sigma^{\frac{1-\gamma}{2}} q_\lambda(\hat{\Sigma}_{Id,\mathcal{A}_1})(\Sigma_{Id,\mathcal{A}_1}^\lambda)^{1/2} \left[ \frac{1}{n} \sum_{i=1}^n (\xi(x_i, y_i)) \right] + \Sigma^{\frac{1-\gamma}{2}} r(\hat{\Sigma}_{Id,\mathcal{A}_1}) g_\lambda$$

$$= \underbrace{\Sigma^{\frac{1-\gamma}{2}} q_\lambda(\hat{\Sigma}_{Id,\mathcal{A}_1})(\Sigma_{Id,\mathcal{A}_1}^\lambda)^{1/2} \left[ \frac{1}{n} \sum_{i=1}^n (\xi(x_i, y_i) - \mathbb{E}_P \xi(x, y)) \right]}_{(I)}$$

$$+ \underbrace{\Sigma^{\frac{1-\gamma}{2}} q_\lambda(\hat{\Sigma}_{Id,\mathcal{A}_1})(\Sigma_{Id,\mathcal{A}_1}^\lambda)^{1/2} \mathbb{E}_P \xi(x, y)}_{(II)} + \underbrace{\Sigma^{\frac{1-\gamma}{2}} r(\hat{\Sigma}_{Id,\mathcal{A}_1}) g_\lambda}_{(III)},$$

(17)

where we take the random variable $\xi(x, y)$ as $\xi(x, y) = (\Sigma_{Id,\mathcal{A}_1} + \lambda)^{-1/2}(y\mathcal{A}_2 K_x - \mathcal{A}_1 g_\lambda(x) K_x)$ which satisfies $\mathbb{E}_Q \xi_2 = (\Sigma_{Id,\mathcal{A}_1} + \lambda)^{-1/2}(\mathcal{A}_2 f_Q - \Sigma_{Id,\mathcal{A}_1}^Q g_\lambda)$ where $f_Q = \mathbb{E}_Q f(x) K_x$ and $\Sigma_{Id,\mathcal{A}_1}^Q = \mathbb{E}_Q K_x \otimes \mathcal{A}_1 K_x$ for arbitrary distribution $Q$ and $\mathbb{E}_P \xi(x, y) = (\Sigma_{Id,\mathcal{A}_1} + \lambda)^{-1/2}(\mathcal{A}_2 S^* f_\rho - \Sigma_{Id,\mathcal{A}_1} g_\lambda)$. We bound different terms (I), (II) and (III) separately and combine them to get the final upper bound. We show that (I) is the mean variance term and is at the scale $\frac{\mathcal{N}(\lambda)}{n} = \frac{\mathrm{Tr}(\Sigma_{Id,\mathcal{A}_1} + \lambda)^{-1} \Sigma_{\mathcal{A}_2,\mathcal{A}_2}}{n}$ when the problem is regular. Term (II) and (III) is smaller than the bias. Our bound of term (III) bounds tighter than [18] (the second term, Lemma 10) via the spectral representation.

**Bounding term (I).** The term (I) is the concentration error of the random variable $\xi(x, y)$ and can be bounded via a Bernstein Inequality. We first bound term (I) via the following decomposition

$$\left\| \Sigma^{\frac{1-\gamma}{2}} q_\lambda(\hat{\Sigma}_{Id,\mathcal{A}_1})(\Sigma_{Id,\mathcal{A}_1}^\lambda)^{1/2} \left[ \frac{1}{n} \sum_{i=1}^n (\xi(x_i, y_i) - \mathbb{E}_P \xi(x, y)) \right] \right\|_H^2 \leq \| \Sigma^{\frac{1-\gamma}{2}} (\Sigma_{Id,\mathcal{A}_1}^\lambda)^{-1/2} \|^2 \left\| \frac{1}{n} \sum_{i=1}^n (\xi(x_i, y_i) - \mathbb{E}_P \xi) \right\|_H^2$$

$$\cdot \| (\Sigma_{Id,\mathcal{A}_1}^\lambda)^{1/2} (\hat{\Sigma}_{Id,\mathcal{A}_1}^\lambda)^{-1/2} \|^2 \| (\hat{\Sigma}_{Id,\mathcal{A}_1}^\lambda)^{1/2} q_\lambda(\hat{\Sigma}_{Id,\mathcal{A}_1})(\hat{\Sigma}_{Id,\mathcal{A}_1}^\lambda)^{1/2} \|^2 \| (\hat{\Sigma}_{Id,\mathcal{A}_1}^\lambda)^{-1/2} (\Sigma_{Id,\mathcal{A}_1}^\lambda)^{1/2} \|^2,$$

where $\Sigma_{Id,\mathcal{A}_1}^\lambda = \Sigma_{Id,\mathcal{A}_1} + \lambda I$ and $\hat{\Sigma}_{Id,\mathcal{A}_1}^\lambda = \hat{\Sigma}_{Id,\mathcal{A}_1} + \lambda I$. At the same time, we knows $\| \Sigma^{\frac{1-\gamma}{2}} (\Sigma_{Id,\mathcal{A}_1} + \lambda)^{-1/2} \|^2 \leq \lambda^{-\frac{\gamma\alpha+p}{\alpha+p}}$ (From lemma B.4) and $\| (\Sigma_{Id,\mathcal{A}_1}^\lambda)^{1/2} (\hat{\Sigma}_{Id,\mathcal{A}_1}^\lambda)^{-1/2} \|^2 \leq 2$ (From lemma B.5) with high probability. At the same time, we have

$$\| (\hat{\Sigma}_{Id,\mathcal{A}_1}^\lambda)^{1/2} q_\lambda(\hat{\Sigma}_{Id,\mathcal{A}_1})(\hat{\Sigma}_{Id,\mathcal{A}_1}^\lambda)^{1/2} \| = \sup_{\sigma \in \sigma(\hat{\Sigma}_{Id,\mathcal{A}_1}^\lambda)} (\sigma + \lambda) q_\lambda(\sigma) \leq 2c_q.$$

Thus we only need to focus on bounding the concentration error $\frac{1}{n} \sum_{i=1}^n (\xi(x_i, y_i) - \mathbb{E}_P \xi)$. We recall the moment condition to control the noise of the observations. There are constants $\sigma, L > 0$ such that

$$\int_\mathbb{R} |y - f^*(x)|^m P(dy|x) \leq \frac{1}{2} m! \sigma^2 L^{m-2}$$

is satisfied for $\mu$-almost all $x \in X$ and all $m > 2$. Note that the moment condition is satisfied for Gaussian noise with bounded variance or have a bounded observation noise. Then we can bound the second order momentum of the random variable $\xi(x,y) = (\Sigma_{Id,\mathcal{A}_1} + \lambda)^{-1/2}(y\mathcal{A}_2K_x - \mathcal{A}_1g_\lambda(x)K_x)$ via decomposing the random into three parts $(\Sigma_{Id,\mathcal{A}_1} + \lambda)^{-1/2}(y\mathcal{A}_2K_x - f^*(x)\mathcal{A}_2K_x)$, $(\Sigma_{Id,\mathcal{A}_1} + \lambda)^{-1/2}(f^*(x)\mathcal{A}_2K_x - \mathcal{A}_2f^*(x)K_x)$ and $(\Sigma_{Id,\mathcal{A}_1} + \lambda)^{-1/2}(\mathcal{A}_2f^*(x)K_x - \mathcal{A}_1g_\lambda(x)K_x)$. Base on the decomposition, we can bound the moments of random variable $\xi(x,y)$ as

$$
\begin{aligned}
\mathbb{E}_P\|\xi(x,y)\|_H^m &= \int \left[ \|(\Sigma_{Id,\mathcal{A}_1} + \lambda)^{-1/2}\mathcal{A}_2K_x\|_H^m \int_{\mathbb{R}} |y - f^*(x)|^m P(dy|x) \right] + \int \left[ \|(\Sigma_{Id,\mathcal{A}_1} + \lambda)^{-1/2}\mathcal{A}_2K_x\|_H^m \|f\|_\infty^m \right] \\
&\quad + \int \left[ \|(\Sigma_{Id,\mathcal{A}_1} + \lambda)^{-1/2}K_x\|_H^m \int_{\mathbb{R}} |\mathcal{A}_2f^*(x) - \mathcal{A}_1g_\lambda|^m P(dy|x) \right] dv(x) \\
&\leq \frac{1}{2}m!\sigma^2 (L + \|f\|_\mathcal{H})^m \|h_x^1\|_\mathcal{H}^{m-2}\text{trace}((\Sigma_{Id,\mathcal{A}_1} + \lambda)^{-1}\Sigma_{\mathcal{A}_2,\mathcal{A}_2}) \\
&\quad + \|h_x^2\|_\mathcal{H}^{m-2}\|\mathcal{A}_2f^*(x) - \mathcal{A}_1g_\lambda\|_{L_\infty}^{m-2} \int |\mathcal{A}_2f^*(x) - \mathcal{A}_1g_\lambda|^2 d\mu(x) \\
&\lesssim m! \left( \|h_x^1\| \right)^m \left[ \sigma^2\text{trace}((\Sigma_{Id,\mathcal{A}_1} + \lambda)^{-1}\Sigma_{\mathcal{A}_2,\mathcal{A}_2}) \right] + m! \left( L_\lambda\|h_x^2\| \right)^{m-2} \left[ \|h_x^2\|^2\|\mathcal{A}_2f^*(x) - \mathcal{A}_1g_\lambda\|_2^2 \right]
\end{aligned}
$$

where $L_\lambda = \|\mathcal{A}_2f^*(x) - \mathcal{A}_1g_\lambda\|_{L_\infty}$, $h_x^1 = (\Sigma_{Id,\mathcal{A}_1} + \lambda)^{-1/2}\mathcal{A}_2K_x$ and $h_x^2 = (\Sigma_{Id,\mathcal{A}_1} + \lambda)^{-1/2}K_x$. The two vectors' norms are bounded in Lemma B.3 as $\mathcal{N}_\infty^1(\lambda) = \sup_{x\in\rho(x)} \|(\Sigma_{Id,\mathcal{A}_1} + \lambda)^{-1/2}K_x\|_H^2 \leq \|k_v^\alpha\|_\infty^2 \lambda^{-\frac{\mu\alpha+p}{\alpha+p}}$, and $\mathcal{N}_\infty^2(\lambda) = \sup_{x\in\rho(x)} \|(\Sigma_{Id,\mathcal{A}_1} + \lambda)^{-1/2}K_x\|_H^2 \leq \|k_v^\alpha\|_\infty^2 \lambda^{-\frac{\mu\alpha+p+2q}{\alpha+p}}$. At the same time, we know that $\|\mathcal{A}_1g_\lambda - \mathcal{A}_2f_\rho\|_2 \lesssim \lambda^{\frac{\beta\alpha-2p}{2(\alpha+p)}} \|u^*\|_{[H]^\beta}$ from Lemma B.9 and $\text{Trace}((\Sigma_{\mathcal{A}_1} + \lambda)^{-1}\Sigma_{Id,Id}) \leq D\lambda^{\frac{p-1}{p+\alpha}}$ from Lemma B.2. Then using Bernstein Inequality (Theorem B.6), we knows that with probability $1 - 2e^{-\tau}$

$$
\begin{aligned}
&\|\frac{1}{n}\sum_{i=1}^n (\xi(x_i,y_i) - \mathbb{E}_P\xi(x,y))\|_H^2 \\
&\lesssim \frac{32\tau^2}{n} \left( \sigma^2\text{trace}((\Sigma_{Id,\mathcal{A}_1} + \lambda)^{-1}\Sigma_{\mathcal{A}_2,\mathcal{A}_2}) + \|h_x^2\|^2\|\mathcal{A}_2f^*(x) - \mathcal{A}_1g_\lambda\|_2^2 + \frac{L_\lambda\|h_x^2\| + \|h_x^1\|}{n} \right) \\
&\lesssim \frac{\tau^2}{n} \left( \sigma^2(\lambda)^{-\frac{1}{p+\alpha} - \frac{p-2q}{p+\alpha}} + \lambda^{-\frac{\mu\alpha-p}{\alpha+p}}\lambda^{\frac{\alpha\beta-2p}{\alpha+p}} + \frac{L_\lambda\|h_x^2\| + \|h_x^1\|}{n} \right)
\end{aligned}
$$
$$\tag{18}$$

Thus we have the final bound $\left\| \Sigma^{\frac{1-\gamma}{2}}q_\lambda(\hat{\Sigma}_{Id,\mathcal{A}_1})(\Sigma_{Id,\mathcal{A}_1}^\lambda)^{1/2} \left[ \frac{1}{n}\sum_{i=1}^n (\xi(x_i,y_i) - \mathbb{E}_P\xi(x,y)) \right] \right\|_H^2 \leq \frac{\tau^2}{n}\lambda^{-\frac{\gamma\alpha+p}{\alpha+p}} \left( \sigma^2(\lambda)^{-\frac{1}{p+\alpha} - \frac{p-2q}{p+\alpha}} + \lambda^{-\frac{\mu\alpha-p}{\alpha+p}}\lambda^{\frac{\alpha\beta-2p}{\alpha+p}} + \frac{L_\lambda\|h_x^2\| + \|h_x^1\|}{n} \right)$.

*Remark* 2. In this remark, we'll bound the $L_\lambda = \|\mathcal{A}_2f^*(x) - \mathcal{A}_1g_\lambda\|_{L_\infty}$ here. For the embedding theorem of the $\ell_\infty$, $L_\lambda \leq \|\mathcal{A}_2f^*(x) - \mathcal{A}_1g_\lambda\|_\mu \lesssim \lambda^{-\frac{(\mu-\beta)_+ + \alpha}{\alpha+p}}$. From Lemma B.3, we know that $\|h_x^1\|_H^2 \lesssim \lambda^{-\frac{\mu\alpha+p+2q}{\alpha+p}}$ and $\|h_x^2\|_H^2 \lesssim \lambda^{-\frac{\mu\alpha+p}{\alpha+p}}$.

**Bounding term (III).** At last we bound the term $\Sigma^{\frac{1-\gamma}{2}}r(\hat{\Sigma}_{Id,\mathcal{A}_1})g_\lambda$ via the following decomposition

$$
\begin{aligned}
\|\Sigma^{\frac{1-\gamma}{2}}r(\hat{\Sigma}_{Id,\mathcal{A}_1})g_\lambda\|_\mathcal{H} &= \|\Sigma^{\frac{1-\gamma}{2}} \underbrace{(\Sigma_{Id,\mathcal{A}_1}^\lambda)^{-1/2}(\Sigma_{Id,\mathcal{A}_1}^\lambda)^{1/2}}_{Id} \underbrace{(\hat{\Sigma}_{Id,\mathcal{A}_1}^\lambda)^{-1/2}(\hat{\Sigma}_{Id,\mathcal{A}_1}^\lambda)^{1/2}}_{Id} r(\hat{\Sigma}_{Id,\mathcal{A}_1}^\lambda)(\Sigma_{Id,\mathcal{A}_1}^\lambda)^{-1}\mathcal{A}_2S^*f_\rho\|_\mathcal{H} \\
&\leq \|\Sigma^{\frac{1-\gamma}{2}}(\Sigma_{Id,\mathcal{A}_1}^\lambda)^{-1/2}\|\|(\Sigma_{Id,\mathcal{A}_1})^{1/2}(\hat{\Sigma}_{Id,\mathcal{A}_1})^{-1/2}\|\|(\hat{\Sigma}_{Id,\mathcal{A}_1})^{1/2}r(\hat{\Sigma}_{Id,\mathcal{A}_1})\|\|(\Sigma_{Id,\mathcal{A}_1}^\lambda)^{-1}\mathcal{A}_2S^*f_\rho\|,
\end{aligned}
$$

where we use $\Sigma_{Id,\mathcal{A}_1}^\lambda$ to denote $\Sigma_{Id,\mathcal{A}_1} + \lambda I$. From Lemma B.4 we now that $\|\Sigma^{\frac{1-\gamma}{2}}(\Sigma_{Id,\mathcal{A}_1}^\lambda)^{-1/2}\|^2 \leq \lambda^{-\frac{\gamma\alpha+p}{\alpha+p}}$. Then we bound the term $\|\Sigma^{\frac{1-\gamma}{2}}r(\hat{\Sigma}_{Id,\mathcal{A}_1})g_\lambda\|_\mathcal{H}$ using $r_\lambda(x)x^u \lesssim \lambda^u$ and get

$$\|(\hat{\Sigma}_{Id,\mathcal{A}_1})^{1/2}r(\hat{\Sigma}_{Id,\mathcal{A}_1})\| = \sup_{\sigma \in \sigma(\hat{\Sigma}_{Id,\mathcal{A}_1}^\lambda)} (\sigma + \lambda)^{1/2}r_\lambda(\sigma) \le \lambda^{1/2}. \tag{19}$$

At the same time, we can bound $\|(\Sigma_{Id,\mathcal{A}_1}^\lambda)^{-1}\mathcal{A}_2 S^* f_\rho\|$ using the spectral representation

$$\|(\Sigma_{Id,\mathcal{A}_1}^\lambda)^{-1}\mathcal{A}_2 S^* f_\rho\|^2 = \sum_{i=1}^\infty \frac{\mu_i p_i^2 a_i^2}{(\lambda + \mu_i p_i)^2} \lesssim \left(\sup_{i \ge 1} \frac{\mu_i p_i^2 \mu_i^\beta}{(\lambda + \mu_i p_i)^2}\right) \sum_{i \ge 1} \mu_i^{-\beta} a_i^2$$

$$\le \left(\lambda^{\frac{(\frac{1-\beta}{2})\alpha+p}{\alpha+p}-1}\right)^2 \|u^*\|_{[H]^\beta}^2 \le \lambda^{\frac{\beta}{\alpha+p}-1}\|u^*\|_{[H]^\beta}^2 \tag{20}$$

Thus we know that

$$\|(\hat{\Sigma}_{Id,\mathcal{A}_1})^{1/2}r(\hat{\Sigma}_{Id,\mathcal{A}_1})\| \lesssim \lambda^{-\frac{\gamma\alpha+p}{2(\alpha+p)}}\lambda^{1/2}\lambda^{\frac{\beta}{2(\alpha+p)}-\frac{1}{2}} \le \lambda^{-\frac{(\beta-\gamma)\alpha}{2(\alpha+p)}},$$

where the last inequality is because $p < 0$ in our assumption.

**Bounding term (II).** In this paragraph, we demonstrate the proof to bound the term $\Sigma^{\frac{1-\gamma}{2}}q_\lambda(\hat{\Sigma}_{Id,\mathcal{A}_1})(\Sigma_{Id,\mathcal{A}_1}^\lambda)^{1/2}\mathbb{E}_P\xi(x,y) = \Sigma^{\frac{1-\gamma}{2}}q_\lambda(\Sigma_{Id,\mathcal{A}_1}^\lambda)^{1/2}(\hat{\Sigma}_{Id,\mathcal{A}_1})(\Sigma_{Id,\mathcal{A}_1} + \lambda)^{-1/2}(\mathcal{A}_2 S^* f_\rho - \Sigma_{Id,\mathcal{A}_1}g_\lambda)$. Note that $\Sigma_{Id,\mathcal{A}_1}(\Sigma_{Id,\mathcal{A}_1}^\lambda)^{-1} = I - \lambda(\Sigma_{Id,\mathcal{A}_1}^\lambda)^{-1}$, thus we knows that $\Sigma^{\frac{1-\gamma}{2}}q_\lambda(\hat{\Sigma}_{Id,\mathcal{A}_1})(\Sigma_{Id,\mathcal{A}_1}^\lambda)^{1/2}(\Sigma_{Id,\mathcal{A}_1} + \lambda)^{-1/2}(\mathcal{A}_2 S^* f_\rho - \Sigma_{Id,\mathcal{A}_1}g_\lambda) = \lambda\Sigma^{\frac{1-\gamma}{2}}q_\lambda(\hat{\Sigma}_{Id,\mathcal{A}_1})(\Sigma_{Id,\mathcal{A}_1}^\lambda)^{-1}\mathcal{A}_2 S^* f_\rho$. At the same time, according to our assumption on the spectral filter $q_\lambda$, we know that

$$\|(\Sigma_{Id,\mathcal{A}_1}^\lambda)^{1/2}q(\hat{\Sigma}_{Id,\mathcal{A}_1}^\lambda)(\Sigma_{Id,\mathcal{A}_1}^\lambda)^{1/2}\| = \sup_{\sigma \in \sigma(\hat{\Sigma}_{Id,\mathcal{A}_1}^\lambda)} (\sigma + \lambda)q_\lambda(\sigma) \le 2c_q.$$

Thus we can bound $\Sigma^{\frac{1-\gamma}{2}}q_\lambda(\hat{\Sigma}_{Id,\mathcal{A}_1})\mathbb{E}_P\xi(x,y)$ via the following decomposition

$$\|\Sigma^{\frac{1-\gamma}{2}}q_\lambda(\hat{\Sigma}_{Id,\mathcal{A}_1})\mathbb{E}_P\xi(x,y)\| = \lambda\|\Sigma^{\frac{1-\gamma}{2}}q_\lambda(\hat{\Sigma}_{Id,\mathcal{A}_1})(\Sigma_{Id,\mathcal{A}_1}^\lambda)^{-1}\mathcal{A}_2 S^* f_\rho\|$$

$$= \lambda\|\Sigma^{\frac{1-\gamma}{2}}\underbrace{(\Sigma_{Id,\mathcal{A}_1}^\lambda)^{-1/2}(\Sigma_{Id,\mathcal{A}_1}^\lambda)^{1/2}}_{Id}\underbrace{(\hat{\Sigma}_{Id,\mathcal{A}_1}^\lambda)^{-1/2}(\hat{\Sigma}_{Id,\mathcal{A}_1}^\lambda)^{1/2}}_{Id}q(\hat{\Sigma}_{Id,\mathcal{A}_1}^\lambda)(\Sigma_{Id,\mathcal{A}_1}^\lambda)^{1/2}(\Sigma_{Id,\mathcal{A}_1}^\lambda)^{-3/2}\mathcal{A}_2 S^* f_\rho\|_{\mathcal{H}}$$

$$\lesssim \lambda\|\Sigma^{\frac{1-\gamma}{2}}(\Sigma_{Id,\mathcal{A}_1}^\lambda)^{-1/2}\|\|(\Sigma_{Id,\mathcal{A}_1}^\lambda)^{-1}\|\|(\Sigma_{Id,\mathcal{A}_1}^\lambda)^{1/2}\|\|(\Sigma_{Id,\mathcal{A}_1}^\lambda)^{-1}\mathcal{A}_2 S^* f_\rho\|$$

$$\lesssim \lambda\lambda^{-\frac{\gamma\alpha+p}{2(\alpha+p)}}\lambda^{-1}\lambda^{1/2}\lambda^{\frac{\beta}{2(\alpha+p)}-\frac{1}{2}} \le \lambda^{-\frac{(\beta-\gamma)\alpha}{2(\alpha+p)}}$$

The last line is because of $\|\Sigma^{\frac{1-\gamma}{2}}(\Sigma_{Id,\mathcal{A}_1}^\lambda)^{-1/2}\|^2 \le \lambda^{-\frac{\gamma\alpha+p}{\alpha+p}}$ (Lemma B.4), $\|(\Sigma_{Id,\mathcal{A}_1}^\lambda)^{1/2}(\hat{\Sigma}_{Id,\mathcal{A}_1}^\lambda)^{-1/2}\| \le 2$ with high probability (Lemma B.5), $\|(\Sigma_{Id,\mathcal{A}_1}^\lambda)^{-1}\| \le \lambda^{-1}$, $\|(\Sigma_{Id,\mathcal{A}_1}^\lambda)^{-1/2}\mathcal{A}_2 S^* f_\rho\| \le \lambda^{\frac{\beta}{2(\alpha+p)}-\frac{1}{2}}$ (proved while bounding term (III)) and $p < 0$.

## B.2 Final Bound

At this time we can have our final bound in Theorem 3.2 via combining the bound for bias (Appendix B.1.2) and (Appendix B.1.3)

$$\|\hat{q}_\lambda - u^*\|_\gamma^2 \lesssim \|\hat{q}_\lambda - g_\lambda\|_\gamma^2 + \|\hat{q}_\lambda - u^*\|_\gamma^2$$

$$\lesssim \lambda^{\frac{(\beta-\gamma)\alpha}{\alpha+p}} + \frac{\tau^2}{n}\lambda^{-\frac{\gamma\alpha+p}{\alpha+p}}\left(\sigma^2(\lambda)^{-\frac{1}{p+\alpha}-\frac{p-2q}{p+\alpha}} + \lambda^{-\frac{\mu\alpha-p}{\alpha+p}}\lambda^{\frac{\alpha\beta-2p}{\alpha+p}} + \frac{L_\lambda\|h_x^2\| + \|h_x^1\|}{n}\right)$$

$$\lesssim \lambda^{\frac{(\beta-\gamma)\alpha}{\alpha+p}} + \frac{\lambda^{-\frac{\gamma\alpha+2(p-q)+1}{p+\alpha}}}{n} + \frac{\lambda^{\frac{(\beta-\gamma)\alpha-\mu\alpha-p}{\alpha+p}}}{n} + \frac{\lambda^{-\frac{\mu\alpha+p+2q}{\alpha+p}}\lambda^{-\frac{\mu\alpha+p+2q}{\alpha+p}}}{n^2} \tag{21}$$

**Case 1.** $\beta \le \frac{\mu\alpha+2q-p+1}{\alpha}$  In this situation, $\frac{\lambda^{-\frac{\gamma\alpha+2(p-q)+1}{p+\alpha}}}{n}$ is larger than $\frac{\lambda^{\frac{(\beta-\gamma)\alpha-\mu\alpha-p}{\alpha+p}}}{n}$. Thus $\lambda^{\frac{(\beta-\gamma)\alpha}{\alpha+p}} + \frac{\lambda^{-\frac{\gamma\alpha+2(p-q)+1}{p+\alpha}}}{n}$ is the dominating term of the loss upper bound. Thus we can take $\lambda = n^{-\frac{\alpha+p}{\beta\alpha+2(p-q)+1}}$ and leads to $n^{-\frac{(\beta-\gamma)\alpha}{\beta+2(p-q)+1}}$ upper bound. At the same time, the third term is dominated by the second term.

**Case 2.** $\beta > \frac{\mu\alpha+2q-p+1}{\alpha}$ In this situation, $\lambda^{\frac{(\beta-\gamma)\alpha-\mu\alpha-p}{\alpha+p}}{n}$ is larger than $\lambda^{-\frac{\gamma\alpha+2(p-q)+1}{p+\alpha}}{n}$. Thus $\lambda^{\frac{(\beta-\gamma)\alpha}{\alpha+p}} + \frac{\lambda^{\frac{(\beta-\gamma)\alpha-\mu\alpha-p}{\alpha+p}}}{n}$ is the dominating term of the loss upper bound. Thus we can take $\lambda = n^{-\frac{\alpha+p}{\mu\alpha+p}}$ and leads to $n^{-\frac{(\beta-\gamma)\alpha}{\mu\alpha+p}}$ upper bound. At the same time, the third term is also dominated by the second term.

## C  Proof of the Lower Bound

### C.1  Preliminaries on Tools for Lower Bounds

In this section, we repeat the standard tools we use to establish the lower bound. The main tool we use is the Fano's inequality and the Varshamov-Gilber Lemma.

**Lemma C.1** (Fano's methods). *Assume that $V$ is a uniform random variable over set $\mathcal{V}$, then for any Markov chain $V \to X \to \hat{V}$, we always have*

$$\mathcal{P}(\hat{V} \neq V) \geq 1 - \frac{I(V;X) + \log 2}{\log(|\mathcal{V}|)}$$

**Lemma C.2** (Varshamov-Gillbert Lemma,[97] Theorem 2.9). *Let $D \geq 8$. There exists a subset $\mathcal{V} = \{\tau^{(0)}, \cdots, \tau^{(2^{D/8})}\}$ of $D-$dimensional hypercube $\mathcal{H}^D = \{0,1\}^D$ such that $\tau^{(0)} = (0,0,\cdots,0)$ and the $\ell_1$ distance between every two elements is larger than $\frac{D}{8}$*

$$\sum_{l=1}^{D} \|\tau^{(j)} - \tau^{(k)}\|_{\ell_1} \geq \frac{D}{8}, \text{for all } 0 \leq j,k \leq 2^{D/8}$$

### C.2  Proof of the Lower Bound

**Theorem C.3.** *Let $(X, B)$ be a measurable space, $H$ be a separable RKHS on $X$ respect to a bounded and measurable kernel $k$ and operator $\mathcal{A} = (\mathcal{A}_2^{-1}\mathcal{A}_1)$ satisfies Assumption 2.1. We have $n$ random observations $\{(x_i, y_i) \in \mathcal{X} \times \mathcal{Y}\}_{i=1}^n$ of $f^* = \mathcal{A}u, u \in \mathcal{H}^\gamma \cap L_\infty$, i.e. $y_i = f^*(x_i) + \eta_i$ where $\eta_i$ is a random noise satisfies the momentum assumption $\mathbb{E}|\eta|^m \leq \frac{1}{2}m!\sigma^2 L^{m-2}$ for some constant $\sigma, L > 0$. Then for all estimators $H : (\mathcal{X} \times \mathcal{Y})^{\otimes n} \to \mathcal{H}^\gamma$ satisfies*

$$\inf_H \sup_{u^* \in \mathcal{H}^\beta \cap L_\infty} \mathbb{E}\|H(\{(x_i,y_i)\}_{i=1}^n) - u^*\|_\gamma^2 \gtrsim n^{-\frac{(\max\{\beta,\mu\}-\gamma)\alpha}{\max\{\beta,\mu\}\alpha+2(q-p)+1}}$$

*Proof.* To proof the lower bound, we use the standard Fano methods via reducing the lower bound to multiple hypothesis testing. We construct our hypothesis using binary strings $\omega = (\omega_1, \cdots, \omega_m) \in \{0,1\}^m$ ($m$ to be determined later) by defining

$$u_\omega = \left(\frac{\epsilon}{m}\right)^{1/2} \sum_{i=1}^m \omega_i \mu_{i+m}^{\gamma/2} e_{i+m}.$$

If we control $m \lesssim \epsilon^{-\frac{1}{\alpha\beta-\alpha\gamma}}$, then we can always keep $u_\omega \in \mathcal{H}^\beta$ for $\|u_\omega\|_\beta^2 = \frac{\epsilon}{m}\sum_{i=1}^m \omega_i^2 \mu_{i+m}^{-(\beta-\gamma)} \lesssim \epsilon\mu_{2m}^{-(\beta-\gamma)} \lesssim m^{\alpha(\beta-\gamma)}\epsilon = O(1)$. Similarly, we can select $m \lesssim \epsilon^{-\frac{1}{\alpha\mu-\alpha\gamma}}$ to control $\|u_\omega\|_{L^\infty} \leq \|u_\omega\|_\mu \leq O(1)$. At the same time, the associated PDE right hand side function $f_\omega = \mathcal{A}_2^{-1}\mathcal{A}_1 u_\omega = \left(\frac{\epsilon}{m}\right)^{1/2}\sum_{i=1}^m \frac{q_i\omega_i}{p_i}\mu_{i+m}^{\gamma/2}e_{i+m}$.

Using Gilbert-Varshamov Lemma we know that there exists $M \geq 2^{m/8}$ binary strings $\omega^{(1)}, \cdots, \omega^{(k)} \in \{0,1\}^m$ with $\omega^{(0)} = (0,\cdots,0)$ subject to

$$\sum_{i=1}^m \left(\omega_i^{(j)} - \omega_i^{(k)}\right)^2 \geq m/8$$

holds for all $j \neq k$. As consequence, the distance between $f_\omega$ and $f_{\omega'}$ can be lower bounded as $\|u_\omega - u_{\omega'}\|_\gamma^2 = \frac{\epsilon}{m}\sum_{i=1}^m (\omega_i - \omega_i')^2 \geq \epsilon/8$. To apply the Fano method, we still need to bound the mutual information between the uniform distribution over all the hypothesis and the distribution of the observed data. We take $\eta_i$ is sampled form $\mathcal{N}(0, \min\{\sigma, L\}^2)$ which satisfies the momentum

condition. Then we know that this mutual information can be bounded by the following average of KL divergence[97] via

$$I(V, X) = \frac{1}{M_\epsilon} \sum_{j=1}^{M_\epsilon} KL(P_j^n \| P_0^n) = \frac{n}{2\sigma^2 M_\epsilon} \sum_{j=1}^{M_\epsilon} \|f_j - f_0\|_{L_2}^2 \lesssim n\epsilon m_\epsilon^{-\alpha\gamma + 2(p-q)} \qquad (22)$$

Then we apply the Fano's inequality

$$\mathbb{P}(\hat{V} \neq V) \geq 1 - \frac{I(V; X) + \log 2}{\log |V|} = 1 - \frac{\frac{16C^\gamma}{\min\{\sigma, L\}^2} n\epsilon m_\epsilon^{-\alpha\gamma - 2(p-q)} + \log 2}{\frac{\log 2}{8} m_\epsilon}$$

$$= 1 - O\left(n\epsilon\epsilon^{\frac{1+\alpha\gamma + 2(p-q)}{\alpha(\max\{\beta, \mu\} - \gamma)}}\right)$$

Take $\epsilon = n^{-\frac{(\max\{\beta, \mu\} - \gamma)\alpha}{\max\{\beta, \mu\}\alpha + 2(p-q) + 1}}$, we know that with constant probability we have

$$\|H(\{(x_i, y_i)\}_{i=1}^n) - u^*\|_\gamma^2 \gtrsim n^{-\frac{(\max\{\beta, \mu\} - \gamma)\alpha}{\max\{\beta, \mu\}\alpha + 2(p-q) + 1}}$$

$\square$