# OpenReview forum: "Sobolev Acceleration and Statistical Optimality for Learning Elliptic Equations via Gradient Descent"
_NeurIPS.cc/2022/Conference — NeurIPS 2022 Accept_

### Official Review · Reviewer_nVgp · 2022-07-08

**Rating:** 8
**Confidence:** 4
**Soundness:** 4 excellent
**Presentation:** 3 good
**Contribution:** 3 good

**Summary:**

This paper provides a theoretical analysis of the optimization process of the optimization-based PDE solvers. Specifically, the authors (1). first generalize the objective function of PINN and DRM for solving elliptic PDEs into a unified formulation, and (2). then provides upper and lower error bounds for gradient descent over a Reproducing Kernel Hilbert Space using the unified objective function. The discovery is two-fold: (1). In most cases, the upper bound matches the lower bound, which shows in this case gradient descent can achieve statistical optimality. (2). Furthermore, it is also observed that the training time that gradient descent first achieves statistical optimality decreases when higher-order derivatives are employed (called "Sobolev Acceleration"). Numerical experiments are conducted to support the theoretical findings.

**Questions:**

Despite the weakness mentioned above, I have the following two questions:
1. Could you give an intuitive explanation of the Sobolev acceleration, despite the analysis through the form of the achieving time? It will be great if such an explanation can be included in the paper.
2. Could you provide an example of how the eigenvalues of differential operators behave (and how they change when the order of the differential operator gets higher)? This can make the discussion "For differential operators, the p is actually negative (differential operators have large eigenvalues over high-frequency basis)" more clear.

**Limitations:**

Please refer to the "weakness" part. Here I list the typos:

Line 100: $e_j\rightarrow e_i $

Line 106: $k\rightarrow K$, $K_x\rightarrow K_z$

Line 117: $k\rightarrow K$ Line 120: $\sigma\rightarrow \Sigma$

Line 195-197: All $u$ in the inner product should be $theta$ or all $\theta$ in the inner product should be $u$

The first case in Theorem 3.2: it should be $\beta<...$

Line 661-663: do we need the transpose symbol $\top$ here?

Line 671: I guess the $\hat{q}_ {\lambda}$  here is the same as the $\hat{g}_{\lambda}$ in Line 718?

Line 712: $\Sigma_{Id, A_1}$ is not defined.

**Strengths And Weaknesses:**

Strengths:
1. Most importantly, this paper provides a theoretical and sound explanation of an interesting phenomenon (i.e., Sobolev Acceleration) of optimization-based PDE solvers. I believe such a contribution is of the communities' interest.
2. The presentation of this paper is clear and the experiments are detailed.

Weakness:
1. The gradient descent in this paper is actually a modified version. While the authors highlight this and mention "Without our modification, the statistical rate will become sub-optimal in some cases", it will be great if the results with the original gradient descent could be provided to facilitate the understanding of the gap.
2. There are some typos and mismatches of symbols (part of them listed in "limitations"). This paper will benefit from checking the mathematical presentations throughout.

Overall, I am in a positive position for this paper. The paper can be further improved by taking the above suggestions.

---

> ### Author Response · Authors · 2022-08-01
> **Response to Reviewer nVgp**
>
> We are really grateful for  Reviewer nVgp’s insightful comments. In the following, we address your concerns point by point.
>
> Regarding modified gradient descent
> A concentration bound can provide consistency prove. The reason we leave it here is that the variance here will let the bound become suboptimal in some cases. This issue arises in connection to the modified Deep Ritz methods in [1]. The variance of kernel approximation of the Laplacian matrix may become larger than the estimation error. (Detailed discussion see Appendix B in arxiv version of [1]). This can be fixed via a more careful sampling procedure, see the modified Deep Ritz methods in [1] and the semi-supervised learning discussion in [2].
> We tried to discuss the cases when the variance may be optimal and when it may not be. However, the classification of the cases is significantly complicated and cannot be fully exposed cleanly in a single conference paper without presenting the actual results. Thus we leave a detailed discussion of these issues for a future paper that is currently in the works. (The variance term, in this case, is a variance of integral by parts, we aim to analyze this in a separate paper)
>
> Regarding an intuitive explanation of the Sobolev acceleration and eigen-structure of the differential operator.
>
> The implicit acceleration of the Sobolev loss function arises because a differential operator can enlarge the small eigenvalue of the kernel integral operator for high-frequency functions, leading to better condition numbers and faster convergence in these eigenspaces while maintaining the statistical optimality. We’ve explained this in line 34-38. We also recognized that this makes readers hard to find. We’ll emphasize this point in our next version.
>
> The kernel integral operator have  smaller eigenvalues on high-frequency functions but the differential operator have larger eigenvalues on high-frequency functions (as an example, Fourier basis are the Eigen of translation invariance kernel integral operator and differential operators.) This is the reason why p is  always negative
>
>
> Thanks for pointing out the notation mistakes. Sorry for the confusion. We’ve fixed our mistake.
>
>
> [1] Lu Y, Chen H, Lu J, et al. Machine Learning For Elliptic PDEs: Fast Rate Generalization Bound, Neural Scaling Law and Minimax Optimality. ICLR 2022.
>
> [2] Murata T, Suzuki T. Gradient Descent in RKHS with Importance Labeling International Conference on Artificial Intelligence and Statistics. PMLR, 2021: 1981-1989.

---

> > ### Comment · Reviewer_nVgp · 2022-08-06
> > **Satisfactory Reply**
> >
> > I would like to thank the authors for the reply and my concerns are dually addressed. I have increased my score to 8.

---

### Official Review · Reviewer_bLL7 · 2022-07-11

**Rating:** 6
**Confidence:** 2
**Soundness:** 3 good
**Presentation:** 3 good
**Contribution:** 3 good

**Summary:**

This paper considers the statistical optimality of gradient descent for solving a class of inverse problems. When the function of interest is contained in a suitable Reproducing Kernel Hilbert Space, the authors show that the gradient descent of a general objective function achieves statistical optimality. As an implication, the authors show that using a Sobolev norm as the objective function for training leads to an implicit acceleration in terms of training efficiency.

**Questions:**

Please see the questions in the above section.

**Limitations:**

I do not see any negative societal impact.

I believe the quality of this work can be improved if the dependence on the problem dimension is also taken into account. Besides, being able to handle more complicated boundary conditions will also be a big plus.

**Strengths And Weaknesses:**

Strengths:

This paper considers the statistical limitations of solving PDEs by parameterizing the solution as a function in a Reproducing Kernel Hilbert Space. I believe the study of the statistical limitations of solving PDEs with a parameterized model is fundamental and crucial to the literature of "AI for science". The authors show that running gradient descent on a general class of objective function already achieves the statistical optimality and is able to explain the advantage of PINN over DRM via the implicit acceleration of the Sobolev training objective.

Weakness:

1. In lines 54 to 58, the authors provide a nice review of the information theoretical limit for solving PDEs with random samples. To distinguish this submission from the previous works, the authors state that "all these papers assume accessibility of the global solution of empirical loss minimization. In contrast, here we consider the gradient descent algorithm for learning the estimator.". Could you explain why using gradient descent as a specific estimator is of great importance? From a practical perspective, we typically use stochastic gradient for the purpose of training which has quite different behavior from gradient descent. So I guess maybe the major difference lies in the analysis. It would be great if the authors could elaborate on this.
2. I am a little confused by the notation in line 97. Is $f$ related to $f^*$? I thought the goal was to recover $f^*$ as claimed in line 91.
3. What is the meaning of $\mathcal{L}^{\beta/2}$ given that $\mathcal{L}$ is defined in line 111.
4. The definition of the Sobolev norm in Definition 2.2 is not the same as the standard one, could you explain how the standard Sobolev norm can be related or translated to the one in Definition 2.2?
5. I see that the parameters $\alpha, \beta, p, q$ are crucial to the theoretical result in Theorem 3.2. Could you please explain what are the meanings of them?
6. What is the dependence on the dimension $d$? A major problem of previous numerical PDE solvers is that they become inefficient when $d$ is large. Hence, the dependence on $d$ is crucial and should be taken into account.
7. The current submission considers the periodic boundary condition (the torus) and hence the integration over the boundary automatically vanishes. However, this is not true for other more practical boundary conditions like Dirichlet and Neumann boundary conditions. It would be great if the authors could elaborate on this as well. Without this zero boundary integral property, integral by parts will introduce extra terms which might break the proof.
8. The accuracy of in Figure 4(b) seems too low to prove that the solution to the PDE is actually recovered.

---

> ### Author Response · Authors · 2022-08-01
> **Response to Reviewer bLL7 (1/2)**
>
> We are really grateful for  Reviewer bLL7’s comments. In the following, we address your concerns point by point.
>
> Why our problem is important?
> In [3,4], the estimator is an ERM estimator but it doesn’t consider computational feasible estimators. In this paper, we introduced a computationally feasible estimator. Another contribution is that the model in [3,4] is under-parameterized; however, our model has infinity capacity as the number of data points increases. In this case, we should consider implicit/algorithmic regularization (Chapter 9 in [7]) which is an important research topic to understand big models trained by gradient descent. Although the loss landscape may contain infinitely many global minimizers, many of which do not generalize well, in practice our optimizer (e.g. SGD) tends to recover solutions with good generalization properties. Our paper can also be considered as the first implicit regularization paper in “AI for science”.  （i.e. the regularization of early stopped gradient descent is optimal for solving inverse problem.）
>
>
> Yes, stochastic gradient descent is of highly significant importance. But a fundamental step in understanding stochastic gradient descent is first to understand the gradient descent dynamics. The bounds for stochastic gradient descent can be decomposed as a bound for GD and a variance component of the stochastic gradient. [5,6] In our case, the variance of stochastic gradient requires a non-trivial analysis in its own right, which we leave for future research.
>
> Definition of Sobolev space.
> The Sobolev norm definition is a standard one [1,2]. If the kernel is a translation invariant kernel, the mercer’s decomposition tells us that the definition of our Sobolev space is based on the decaying speed of the associated Fourier coefficients, which is equivalent to the standard Sobolev space definition.
>
>
> Regarding notations
> We’ve put most assumptions in Assumption 2.1. Our assumptions and notation are based on standard conventions for kernel regression analysis; see, for example, [1].
> - $\alpha$: The parameter in the capacity condition which represents the decay speed of the Eigen-values of the kernel integral operator  This represents how large the kernel space is.
> - $\beta$: the parameter in the source condition, represents the smoothness of the target function
> - $\mu$: Tthe regularity parameter that the smoothed kernel space can be embedded into L infty space. This represents the l_inf regularity of the kernel.
> The p and q are assumptions we further made for the operator A1 A2, the Eigen “decay” of operator A1 A2. In DRM and PINN, A1 and A2 are all differential operators, so p and q are negative, meaning eigen “increasing”.

---

> > ### Author Response · Authors · 2022-08-01
> > **Response to Reviewer bLL7 (2/2)**
> >
> > Regarding the boundary condition
> > We can have bounds considering the boundary condition but the upper bound will not match the lower bound. However, this gap constitutes an open problem in this area, see [3,4]. A natural program should first consider how to achieve information theoretical optimal bounds for boundary conditions – which is an interesting research problem. (The known bounds for ERM models using DRM and PINN are both not optimal in this case.)
> >
> > Regarding dimension dependence
> > We’ve discussed the dimension dependence in line 228-235, which matches the previous bounds in [3,4]. The rate is similar to the standard nonparametric smoothness/dimension polynomial rates. If the smoothness is a constant proportion of the dimension, one can beat the curse of dimensionality bound. (This type of assumption is almost always needed to beat the curse of dimensionality in learning bounds)
> >
> > Regarding magnitude in figure 4 (b)
> > Different from other ai for pde papers, we fixed the number of points sampled. But all the other comparison papers sample the data online so that the overfitting issue is not really exposed.
> >
> > Minor points
> > in line 97, it should be f* sorry for the confusion. L^beta/2 standard definition by spectral in the kernel literature, we’ll add the definition in the next version
> >
> > [1] Fischer S, Steinwart I. Sobolev Norm Learning Rates for Regularized Least-Squares Algorithms[J]. J. Mach. Learn. Res., 2020, 21: 205:1-205:38.
> >
> > [2] Steinwart I, Christmann A. Support vector machines[M]. Springer Science & Business Media, 2008.
> >
> > [3] Richard Nickl, Sara van de Geer, and Sven Wang. Convergence rates for penalized least  squares estimators in pde constrained regression problems. SIAM/ASA Journal on Uncertainty Quantification, 8(1):374–413, 2020.
> >
> > [4] Yiping Lu, Haoxuan Chen, Jianfeng Lu, Lexing Ying, and Jose Blanchet. Machine learning for elliptic pdes: Fast rate generalization bound, neural scaling law and minimax optimality. arXiv 402 preprint arXiv:2110.06897, 2021.
> >
> > [5] Dieuleveut A, Bach F. Nonparametric stochastic approximation with large step-sizes[J]. The Annals of Statistics, 2016, 44(4): 1363-1399
> >
> > [6] Pillaud-Vivien L, Rudi A, Bach F. Statistical optimality of stochastic gradient descent on hard learning problems through multiple passes[J]. Advances in Neural Information Processing Systems, 2018, 31.
> >
> > [7] https://docs.google.com/viewer?url=https://raw.githubusercontent.com/tengyuma/cs229m_notes/main/master.pdf

---

### Official Review · Reviewer_Ffzb · 2022-07-11

**Rating:** 6
**Confidence:** 1
**Soundness:** 3 good
**Presentation:** 2 fair
**Contribution:** 3 good

**Summary:**

This paper provides the analysis of the statistical limits of the gradient descent algorithm used on a class of objective functions for solving the inverse problem. Specifically, the main use case of this analysis is utilizing neural networks as PDE solvers. The theorem shows in what cases the gradient descent algorithm can achieve an optimal convergence. It also demonstrates the influence of early stopping on PINN and DRM methods.

**Questions:**

Please see weaknesses.

**Limitations:**

The authors have discussed about the limitations.

**Strengths And Weaknesses:**

As far as I understand, utilizing deep learning based numerical methods to solve PDE is a promising direction thus the analysis to understand its convergence is also important. The authors include detailed discussion about the link between the proposed theorem and other existing works, which facilitates the interpretation of the result. The proposed theorems match well with the empirical observation made on PINN and DRM methods about the Sobolev implicit acceleration. One weakness is that the result only applies to gradient descent. If would be more useful if the accelerated gradient descent is covered since it's widely used in practice when training deep networks.

---

> ### Author Response · Authors · 2022-08-01
> **Response to Reviewer Ffzb**
>
> We are really grateful for  Reviewer Ffzb’’s comments and we'are happy to see reviewers consider our paper important.
>
> > accelerated gradient descent
> It’s interesting to consider accelerated gradient descent in our future work and thanks for pointing this out. The acceleration method for the kernel method is still an active research area [1]. It’s interesting to investigate whether Sobolev acceleration can be combined with other acceleration methods.  We believe that this version of the paper already has enough content, especially given the limitations of the conference. But we’ll leave this as future work.
>
> [1] Nicolò Pagliana and Lorenzo Rosasco. Implicit regularization of accelerated methods in Hilbert spaces. arXiv preprint arXiv:1905.13000, 2019.

---

### Official Review · Reviewer_p5b2 · 2022-07-11

**Rating:** 4
**Confidence:** 3
**Soundness:** 3 good
**Presentation:** 4 excellent
**Contribution:** 2 fair

**Summary:**

This paper studies statistical optimality of a slightly modified gradient descent for learning elliptic equations with kernels. This problem is an extension of kernel regression. The first sound theoretical contribution is a min-max lower-bound in Thm.3. Then, the authors analyze the gradient descent on a batch of observations drawn i.i.d. In Thm 3.2. Combining these two results, statistical optimality of gradient is established. The key assumptions providing such statistical optimality relies on the critical capacity condition that holds for ridge regression and also toy examples of PDE solvers. The result is compared to existing analysis in literature which focus on more restrictive instances of the problem such as those in [15,16,68]. Then, the authors provide intuitions on the influence of loss function on the convergence of gradient descent leveraging their theoretical analysis and assumptions.


**Questions:**

1. The authors motivate their study by the example of neural tangent kernel. I wondered their assumptions hold for neural tangent kernels.
2. Would you please elaborate why it is not possible to invoking the existing results on kernel regression (early stopping and the lower-bound analysis) to get the same results?
3. What is the difference between the modified gradient descent in this paper, and gradient descent on kernel regression?
4. Is there a representer theorem for the modified kernel regression? Would you explain more about its connections with the early stopping?

**Strengths And Weaknesses:**

**Strengths**

1. The paper establishes a theoretically sound contribution on min-max lower bound for a generalization of kernel regression.
2. The paper is very well written
3.  Assumptions are stated clearly and motivated by examples.
4. It is interesting to establish the optimality of gradient descent for solving PDEs with kernels.


**Weaknesses**
1.  I am not sure about the significance of the result. Given results from kernel regression, one can translate them to the settings of this paper. Assuming that u(x) and u*(x) are in RKHS with kernel k, then A u(x) = <\theta, A(K_x)> for all operators x. Therefore, given observations $A_2^{-1} A_1 u(x)$, one can compute observations from $A_1 u(x)$ and given that the original measurements are i.i.d, they remain i.i.d.. Then a straightforward change of variables as $y = A^{1/2} u(x)$ leads to the standard kernel regression settings for which all min-max lower bound and the convergence of gradient descent is well known. The only difference with the standard kernel regression setting is the convergence metric which can be sandwiched by the l2 distance in kernel ridge regression settings under standard assumptions. In particular, the change of gradient is equivalent to the change of objective which makes the gradient descent on the modified objective same as the gradient descent on kernel regression.
2. It would be interesting to experimentally compare the performance of gradient descent with its modified variant considered in this paper.

---

> ### Author Response · Authors · 2022-08-01
> **Response to Reviewer tsfh**
>
> We are really grateful for  Reviewer tsfh’s efforts in the comments. We believe that there may be some misunderstandings regarding the setting of our paper, but we will try to clarify them by addressing your concerns point by point. In the following, we address your concerns point by point.
>
> ## Why we can’t inverse A transfer to standard kernel regression? Why standard kernel regression results can’t be used here.
>
> A is an operator that operates on u (i.e. (Au)(x)) but not a matrix that operates on u(x). For example, A can be laplace, since you know Laplace u(x), you can’t directly know u(x). Assuming that inversion of operators is “standard”, would obviate the large and expanding literature on learning PDEs/inverse problems.
>
> One can also first use kernel regression to estimate the function Au, then inverse the A. Firstly, computing the inverse of A involves solving a PDE that is computationally infeasible in high dimension (for one can’t build a grid in high dimension). (see line 40-45) This is why recently machine learning techniques have become popular in numerical PDEs.
>
> Secondly, using this technique (i.e. inverting A), one would need a Sobolev learning rate for early stopped kernel sgd. As far as we know, Sobolev learning rates are so far built only for kernel ridge regression. Our paper provides the first results for Sobolev learning rates for early stopped kernel SGD  (which is a special case of our paper).
>
> Last but not least, we discovered that Sobolev training serves as a preconditioner and leads to faster convergence. As far as the authors know, this is also not observed in the kernel regression literature.
>
> I think this misunderstanding may contribute to the reviewer’s misevaluation of our contribution. We hope that our explanation can help the reviewer better appreciate the contributions and consider re-evaluating our paper in view of this.
>
> ## About Neural Tangent Kernel
> A Neural Tangent Kernel on the sphere satisfies our assumptions for they belong to the family of Laplace kernels, see [1,2], We have cited the two papers and discussed them in lines 240-244.
>
> [1] Alberto Bietti and Francis Bach. Deep equals shallow for relu networks in kernel regimes. arXiv preprint arXiv:2009.14397, 2020.
>
> [2] Lin Chen and Sheng Xu. Deep neural tangent kernel and Laplace kernel have the same rkhs.  arXiv preprint arXiv:2009.10683, 2020.
>
> ## About Representer Theorem
> The representer theorem is already built in [1,2].
>
> [1] Chen Y, Hosseini B, Owhadi H, et al. Solving and learning nonlinear PDEs with Gaussian processes[J]. Journal of Computational Physics, 2021, 447: 110668.
>
> [2] Cabannes V, Pillaud-Vivien L, Bach F, et al. Overcoming the curse of dimensionality with Laplacian regularization in semi-supervised learning[J]. Advances in Neural Information Processing Systems, 2021, 34: 30439-30451.
>
> We hope our response has addressed the concerns raised and that the overall response can help in the reviewer’s understanding of our paper. Based on this, we urge the reviewer to consider improving the evaluation of our paper.

---

> > ### Comment · Reviewer_p5b2 · 2022-08-04
> > **A follow-up question**
> >
> > Thanks for your your detailed response. Consider Theorem 3.1; for this theorem, do we need computing the inverse operator for the proof?

---

> > > ### Author Response · Authors · 2022-08-04
> > > **response**
> > >
> > > We don't include inverse in our algorithm.We use gradient descent as our algorithm which only includes transpose of the operator which is easy to compute. One major part of our proof is to show how this iterative procedure can approximate the inverse.

---

> > > ### Author Response · Authors · 2022-08-04
> > > **follow up**
> > >
> > > Theorem 3.1 is a lower bound. It states that all algorithm can't do better than our bound. No matter how much computational power you have. This is an information theoretical proof

---

> > > > ### Comment · Reviewer_p5b2 · 2022-08-04
> > > > **Theorem 3.1**
> > > >
> > > > Then, your response to the first question is not convincing. My question is why we can not use change of variables (transforming observations) and invoke existing results for kernel regression. Your response is that we can not compute inverse in practice. Still, for theoretical proves we can take the inverse and reduce one problem to another, right? For example, Theorem 3.1. can be obtained by invoking lowerbounds for ridge regression.

---

> > > > > ### Author Response · Authors · 2022-08-04
> > > > > **Regards thm3.1**
> > > > >
> > > > >  The norm of A inverse is not bounded thus you can’t do this. If you check carefully, the rate is different from standard kernel regression rate.

---

> > > > > > ### Comment · Reviewer_p5b2 · 2022-08-04
> > > > > > **Request for more detials**
> > > > > >
> > > > > > It would be great if you exactly cite one of classical result for ridge regression and compare your rate with their minmax rate and explain more details what are your technical novelties in terms of proof and rate. If the proof is based on standard techniques previously used, it would great to provide citations. Why norm $A$ is not bounded? Which norm?
> > > > > >
> > > > > > Consider kernel $K= \sum_{i=1}^n \lambda_i e_i e_i^\top$. According to Assumption 2.1,  $A = \sum_{i} \gamma_i e_i e_i^\top$. Combining these two, we get $A K = \sum_{i} \gamma_i \lambda_i e_i e_i^\top$. Hence $$A u = A \langle \theta, K \rangle = \langle \theta, A K\rangle $$. Therefore, Assumption 2.1 concludes $Au$ lies in another RKHS with kernel $AK$. Now, we can apply classical lower bound for kernel regression with kernel A K to get Theorem 3.1. Please let me know, where I am making mistake.

---

> > > > > > > ### Author Response · Authors · 2022-08-05
> > > > > > > **Regards our contribution**
> > > > > > >
> > > > > > > Firstly, our paper follows traditional kernel regression literature and is cited in the paper as [17,18,42,45-48].
> > > > > > >
> > > > > > > Secondly, the traditional kernel only considers l2 norm convergence. A inverse is not bounded you can't use their result. Once you inverse the A, it's a sobolev norm. Sobolev norm rate convergence is just build in [1], but don't consider the gradinet descent in the paper.
> > > > > > >
> > > > > > > Lastly, combine with [1], your prove may works for (only) **lower bound**. But you still need to translate the source condition, the assumption of A1 and A2 and l_inf regularity condition. However, The original proof is easy, only lies in line 774-785. I don't think your proof can shorten the proof. If we complete this proof, we can also include this proof in our next version.
> > > > > > >
> > > > > > > We still want to defense our contribution. From our perspective, our contribution is mostly build the upper bound, whose proof spans page 16-24. The proof can't be covered by the previous literature and is interesting in the following sense
> > > > > > > - the gd dynamic is different, we considered a more general class of pre-conditioned dynamic
> > > > > > > - we are considering an inverse problem
> > > > > > > - sobolev norm convergence of the gd is still missing in the literature
> > > > > > > - our proof explains why PINN convergence faster then DRM
> > > > > > >
> > > > > > > The lower bound is only build to check our upper bound is optimal. See our contribution summarization in line 73-88. It's unfair to reject a paper because line 774-785 can be proved (still needs a translation) by a recent paper [1]  but missing the contribution of the conclusion proved from page 16-24 (our upper bound and realistic practice suggestion, i.e. PINN convergence faster in iteration number especially low dimension rough problem but DRM is faster for a single iteration).
> > > > > > >
> > > > > > > [1] Fischer S, Steinwart I. Sobolev Norm Learning Rates for Regularized Least-Squares Algorithms[J]. J. Mach. Learn. Res., 2020, 21: 205:1-205:38.

---

> > > > > > > > ### Author Response · Authors · 2022-08-05
> > > > > > > > **Follow up**
> > > > > > > >
> > > > > > > > Have we resolved the reviewer's concern?

---

> ### Author Response · Authors · 2022-08-04
> **Looking forward to the response.**
>
> Please let us know if our response addresses your concerns. We are happy to address any remaining points during the discussion phase. We would like to respectfully emphasize that you have a critical misunderstanding of our work. Due to your high confidence score, please read our responses carefully. We believe our response can clarify all of your concerns which may help you to re-evaluate our contributions.  If our response has adequately addressed your concerns, we kindly ask you to consider raising the score.

---

### Official Review · Reviewer_GamV · 2022-07-13

**Rating:** 6
**Confidence:** 2
**Soundness:** 4 excellent
**Presentation:** 3 good
**Contribution:** 3 good

**Summary:**

The authors study error bounds for estimating the solution $u$ of Poisson’s equation $\Delta u = f$ using early stopped gradient descent. The function $f$ is observed via noisy measurements $y_i$ at random points $x_i$, satisfying $E[y_i | x_i] = f(x_i)$. Poisson’s equation can be solved using the DRM or PINN approaches where the optimization objective function is crafted to be an integral. The integrals are then approximated with empirical risk and then gradient descent can be applied.

The authors establish lower bounds in Sobolev norms, given smoothness levels of $f, u^*$ and the smoothness of the operators $\mathcal{A}_1, \mathcal{A}_2$ that define the integral objective. They further show that early stopped gradient descent can achieve matching upper bounds in several cases and discuss suboptimality in the rest of the cases.


**Questions:**

1. In equation (2), should you remove the $-$ sign to be consistent with the rest of the paper?

2. In line 97, for $\mathcal{A} = \Delta$, $\mathcal{A}$ is not invertible. Should the usage of $\mathcal{A}^{-1}$ be clarified here?

3. Lines 91-92 and line 97: f and u seem to be interchanged, causing some confusion.

4. The relation between $\mathcal{A}, \mathcal{A}_1, \mathcal{A}_2$ in line 94 and line 186 are consistent given the Assumption 2.1 (d), but does not seem to hold in general?

5. Can you please write out the gradient step for DRM and PINN separately in elementary terms, perhaps in the supplement?


**Limitations:**

None.

**Strengths And Weaknesses:**

The results are interesting. The upper bound proof technique is interesting. The lower bound proof is standard, but that is not a negative.

The paper is pretty dense in terms of mathematics. One simplification could be to give examples (such as DRM, PINN) when introducing abstract concepts such as the operators $\mathcal{A}, \mathcal{A}_i$. In general, I suggest that the authors should make the text more amenable to machine learning graduate students.

---

> ### Author Response · Authors · 2022-08-01
> **Response to Reviewer GamV**
>
> We are really grateful for Reviewer GamV’s insightful comments. In the following, we address your concerns point by point.
>
> ## Is laplacian invertible
> Sorry for the confusion, this imprecision can be easily fixed in two ways. The first way considers $\Delta +\lambda I$ instead of $\Delta$, as in [1]. (This still satisfies our assumption for we just consider the speed of the eigenvalue increasing.)  The second way considers that $\Delta$ is invertible under appropriate boundary conditions, for example, assuming that the domain of $Delta$ is restricted to a function space with zero boundary conditions on a regular domain. The third way is to define the pesudo-inverse in the spectral space. The analysis remains largely intact, after adding these clarifications. We’ll update this in the following version of the draft. Thanks for pointing this out and sorry for the confusion in this version.
>
> [1] Lu Y, Chen H, Lu J, et al. Machine learning for elliptic PDEs: fast rate generalization bound, neural scaling law and minimax optimality arXiv preprint arXiv:2110.06897, 2021.
>
> ## The - sign
> It depends on how the Laplacian is defined, we’ll clarify the notation in the final version
>
> ## simplification of the dense math
> Thanks for the suggestion. We’ve reorganized the examples in Section 2.1 and discussed how our model relates to them in lines 169-170. We understand that this may only partially address the issue of a dense exposition in explaining these relationships, but due to the page limit, it’s the best we can do in the submitted version. We’ll try more explanations in the appendix.
>
> ## relationship between A A1 A2
> We are not sure what exactly is the referee asking. We believe that the referee is asking if defining A is necessary, given that some of the assumptions may be stated in terms of A1 and A2 only. If this is the issue, we believe that introducing A is necessary both in order to define the problem and provide the bound comparisons in line 267-264.
>
> ## about f and u
> Sorry for the confusion, we’ve fixed the problem.

---

> > ### Comment · Reviewer_GamV · 2022-08-08
> > **Thanks to the authors.**
> >
> > Thanks for addressing my questions.

---

### Meta-Review · Area_Chair_QXQ9 · 2022-08-27

**Recommendation:** Accept
**Confidence:** Certain

**Metareview:**

This paper studies the statistical performance of Deep Ritz Methods (DRM) and Physics Informed Neural Networks (PINN) for solving an inverse problem with respect to elliptic equations. As a key example, consider the problem of estimating the solution $u$ to Poisson's equation $\Delta u = f$ given noisy observations of $f$. When the solution is a function in an RKHS, and using a Sobolev norm as the objective function, they show that gradient descent achieves implicit acceleration. They use this theory to show that both DRM and PINN achieve statistical optimality, but the number of epochs needed for DRM is larger. This is a nice contribution to the growing area of deep PDE solvers, and provides some illuminating theoretical insights. There was some debate between a reviewer and the authors about whether the results follow from existing bounds for kernel regression, but the authors convincingly argued that the norm of $A^{-1}$ in general need not be bounded and so those results cannot be applied in a blackbox manner. Additionally there are other differences and complications.

**Award:**

No

---

### Decision · Program_Chairs · 2022-09-14

Accept